# On Efficient Bayesian Exploration in Model-Based Reinforcement Learning

**Alberto Caron**                                                    *acaron@turing.ac.uk*
*The Alan Turing Institute*
*London, UK*

**Chris Hicks**                                                    *c.hicks@turing.ac.uk*
*The Alan Turing Institute*
*London, UK*

**Vasilios Mavroudis**                                                    *vmavroudis@turing.ac.uk*
*The Alan Turing Institute*
*London, UK*

**Reviewed on OpenReview:** *https://openreview.net/forum?id=NaO2hDWqkF*

## Abstract

In this work, we address the challenge of data-efficient exploration in reinforcement learning by examining existing principled, information-theoretic approaches to intrinsic motivation. Specifically, we focus on a class of exploration bonuses that targets epistemic uncertainty rather than the aleatoric noise inherent in the environment. We prove that these bonuses naturally signal epistemic information gains and converge to zero once the agent becomes sufficiently certain about the environment's dynamics and rewards, thereby aligning exploration with genuine knowledge gaps. Our analysis provides formal guarantees for IG-based approaches, which previously lacked theoretical grounding. To enable practical use, we also discuss tractable approximations via sparse variational Gaussian Processes, Deep Kernels and Deep Ensemble models. We then outline a general framework — Predictive Trajectory Sampling with Bayesian Exploration (PTS-BE) — which integrates model-based planning with information-theoretic bonuses to achieve sample-efficient deep exploration. We empirically demonstrate that PTS-BE substantially outperforms other baselines across a variety of environments characterized by sparse rewards and/or purely exploratory tasks.

## 1 Introduction

The exploration-exploitation trade-off is a long-standing challenge in Reinforcement Learning (RL) (Sutton & Barto, 2018). Exploration in classic RL algorithms is often achieved via simple heuristics such as $\epsilon$-greedy policy in Q-methods (Mnih et al., 2015), action noise injection (Lillicrap et al., 2015), or some form of policy entropy regularizers in policy gradient methods (Sutton et al., 1999; Kakade, 2001; Schulman et al., 2017). In some environments, these simple heuristics are enough to ensure sufficient exploration under uncertainty such that the optimal policy can eventually be learned; in others, characterized by more 'sparse' rewards and noisy transitions, they are prone to get the agent stuck in sub-optimal policy regions instead.

Different methods that satisfy the need for deeper exploration in sparse rewards environments have been proposed, e.g., Thompson sampling adaptations (Osband et al., 2016; 2018) and Q-network parameter uncertainty (Azizzadenesheli et al., 2018). Perhaps the most popular class of methods is the one based on *intrinsic motivation* (Schmidhuber, 1991; Chentanez et al., 2004), or *intrinsic rewards.* Intrinsic motivation methods introduce an additional reward signal that encourages the agent to explore and learn about aspects of the environment beyond what the task-specific rewards alone can incentivize (Ladosz et al., 2022). This means that when the agent encounters a new state, they assign it a higher internal, or 'intrinsic', exploration

bonus $r_t^i$ that encourages them to visit that state more often. Several different types of exploration bonuses have been proposed, including *model-based* (Stadie et al., 2015; Osband et al., 2016; Pathak et al., 2017), where the agent learns a model of the environment dynamics and uses an output from it (prediction error, variance, etc.) to define $r_t^i$, and *count-based* (Bellemare et al., 2016; Ostrovski et al., 2017; Tang et al., 2017), where the intrinsic reward is defined as a function of the visitation frequency of a given state-action pair.

However, some intrinsic reward methods may inadvertently conflate *epistemic* uncertainty (lack of knowledge about the environment due to limited data) with *aleatoric* uncertainty (inherent stochasticity that cannot be reduced through data collection) (Hüllermeier & Waegeman, 2021). For instance, in highly noisy environments, exploration methods based on prediction error may keep rewarding the agent for visiting already well-understood states whenever they produce random transitions or outcomes. As a consequence, the agent may continue exploring those regions indefinitely, rather than shifting its exploration budget to truly under-explored regions, resulting in high sample inefficiency. This challenge has led researchers to begin to consider alternative intrinsic rewards specifications that attempt to "de-noise" the transitions (Pathak et al., 2017; Jarrett et al., 2023), looking for underlying novelty signals. The overall aim of this paper is to provide theoretically grounded insights for constructing good exploration measures that are naturally capable of dissecting aleatoric and epistemic components.

**Related Work**  Model-based intrinsic rewards (Bellemare et al., 2016; Osband et al., 2016; Pathak et al., 2017) are a popular class of methods that address sparse rewards by leveraging exploration bonuses derived from a learned model of the dynamics. Examples include next-state prediction error (Stadie et al., 2015; Pathak et al., 2017) and variance (Sorg et al., 2010; Pathak et al., 2019). Information-theoretic intrinsic rewards, rooted in Bayesian Optimal Experimental Design (BOED) (Lindley, 1956; Pukelsheim, 2006; Foster et al., 2019; Rainforth et al., 2024) and Bayesian Active Learning (BAL) (Houlsby et al., 2011; Hanneke et al., 2014; Gal et al., 2017) principles, have been previously proposed. Notably, Houthooft et al. (2016) introduced Variational Information Maximizing Exploration (VIME), defining intrinsic rewards using Bayesian surprise (Itti & Baldi, 2009), quantified as Information Gain (IG) via relative entropy between posterior and prior parameters. VIME specifically computes a one-step IG bonus using a variational Bayesian neural network and incorporates this within a model-free policy-gradient learner. More recently, few contributions proposed employing IG within a model-based planning framework (mostly via the use of deep ensembles specifically as a forward dynamics model) that explicitly considers longer planning horizons (Shyam et al., 2019; Sekar et al., 2020). However, none of these prior contributions provide a formal theoretical analysis of the exploration bonuses proposed. Tangentially related is the literature on Model-Based RL (MBRL) frameworks that have frequently leveraged uncertainty quantification for exploration purposes, such as Gaussian Process-based planners (Deisenroth & Rasmussen, 2011; Kamthe & Deisenroth, 2018; Mehta et al., 2022). Yet, many of these works either focus solely on dynamics uncertainty or do not explicitly address sparse reward environments or purely exploratory tasks. While the idea of integrating information gain bonuses within MBRL planners is present in prior work, formal analyses characterizing their theoretical properties — such as convergence and contraction rates — remain largely unexplored. Finally, our work also connects closely to the seminal literature on Bayesian RL (Strens, 2000; Duff, 2002; Ghavamzadeh et al., 2015), which inherently balance exploration and exploitation through Bayesian uncertainty quantification.

**Contributions**  In this work, we study a class of information-theoretic exploration bonuses inspired by Bayesian surprise and rigorously analyze their theoretical and practical properties. Although employing information gain itself as an exploration bonus is not new, our contributions are as follows: i) we provide a rigorous theoretical foundation for information-theoretic exploration bonuses, proving key convergence and contraction properties, which nicely complement prior works such as VIME (Houthooft et al., 2016) and other existing "planning-to-explore" approaches (Shyam et al., 2019; Sekar et al., 2020), which lack such formal analysis; ii) we advocate for the use of model-based planning approaches for settings that require high data efficiency, and design a general Predictive Trajectory Sampling with Bayesian Exploration (PTS-BE) framework, which simply generalizes planning-to-explore methods by supporting arbitrary Bayesian models (GPs, deep ensembles, etc.), and explicitly model epistemic uncertainty both in next state and rewards predictions; iii) we provide an empirical comparison of different tractable Bayesian approximators for posterior inference — including deep ensembles, Gaussian Processes, and Deep Kernels — and analyze their impact

on information gain estimation and exploration efficiency; iv) we empirically demonstrate the theoretical properties of these information-theoretic bonuses and thoroughly investigate their practical benefits in terms of sample efficiency when combined with standard policy-gradient algorithms (e.g., Proximal Policy Optimization, Soft Actor-Critic), particularly emphasizing their effectiveness in sparse-reward environments and exploratory tasks that inherently demand careful uncertainty quantification and strategic exploration. Overall, this work is intended to provide novel theoretical insights into the properties of Bayesian information-based intrinsic rewards and to introduce a generalized and theoretically justified MBRL planning-to-explore framework.

## 2    Problem Setup

We begin by defining some concepts and notation that will be used throughout the paper. Firstly, consider the canonical definition of a **Markov Decision Processes** (MDP) (Puterman, 2014; Sutton & Barto, 2018). A MDP is a tuple $\langle \mathcal{S}, \mathcal{A}, p_\tau, p_0, r^e, \gamma \rangle$, defined over an horizon of $t \in \{1, ..., T\}$ time steps where: i) $\mathcal{S}$ is the state space; ii) $\mathcal{A}$ is the action space; iii) $p_\tau(\cdot|s,a) \in \mathcal{P}_\tau$ is the state transition probability that determines the next state $s_{t+1}$ given the current state and action pair $(s_t, a_t)$; iv) $p_0 \in \mathcal{P}_0$ is the initial state probability, $s_0 \sim p_0(\cdot)$; v) $r^e : \mathcal{S} \times \mathcal{A} \to \mathcal{C} \subset \mathbb{R}$ is a bounded *extrinsic* reward function output from the environment; vi) $\gamma \in (0,1)$ is a discount factor.

A policy is a function $\pi : \mathcal{S} \to \mathcal{A}$ that defines an agent's behaviour by mapping a certain state $s \in \mathcal{S}$ to an action $a \in \mathcal{A}$, and can be either deterministic or stochastic. Given a certain policy $\pi \equiv \pi(a_t|s_t)$ and transition probability $p_\tau \equiv p(s'|s,a)$, the action-value function $Q^{\pi,p_\tau}(s,a)$ can be defined via the one-step Bellman equation, $Q^{\pi,p_\tau}(s,a) = r^e(s,a) + \gamma \mathbb{E}_{p_\tau}[V^{\pi,p_\tau}(s')]$, where $V^{\pi,p_\tau}(s)$ is the state-value function instead: $V^{\pi,p_\tau}(s) = \mathbb{E}_{a\sim\pi(\cdot|s)}[r^e(s,a) + \gamma\,\mathbb{E}_{p_\tau}[V^{\pi,p_\tau}(s')]]$. Lastly, another useful quantity is the discounted returns defined as $J(\pi;p_\tau) = \sum_{t=0}^{T} \gamma^t \mathbb{E}_{a_t\sim\pi,s'\sim p_\tau}[r^e(s_t,a_t)]$. An optimal policy $\pi^*$ is equivalently defined as the maximizer of $Q^{\pi,p_\tau}(s,a)$, $V^{\pi,p_\tau}(s)$ and $J(\pi;p_\tau)$, for all $s \in \mathcal{S}$.

### 2.1    Intrinsic Curiosity and Exploration Bonuses

Intrinsic curiosity, or intrinsic motivation, (Schmidhuber, 1991; Barto, 2013) in RL is a technique that involves augmenting the environment's *extrinsic* reward signal $r^e(s_t,a_t)$ with some notion of *intrinsic* reward $r_t^i(s_t,a_t) \in \mathbb{R}$, such that the total reward signal at time $t$ from pair $(s_t,a_t)$ received by the agent equals $r(s_t,a_t) = r^e(s_t,a_t) + \eta_t r^i(s_t,a_t)$, where $\eta_t$ is a scaling factor that can be decayed as a function of the time step $t$. The intrinsic reward $r_t^i \equiv r^i(s_t,a_t)$ constitutes a form of exploration bonus that the agent gets to enhance visitation of unseen regions of the state-action space $\mathcal{S}\times\mathcal{A}$. This intrinsic reward augmentation technique, which generates the cumulative sum of discounted returns $J(\pi;p_\tau) = \sum_{t=0}^{T} \gamma^t \mathbb{E}_{a_t\sim\pi,s'\sim p_\tau}[r^e(s_t,a_t) + \eta\,r^i(s_t,a_t)]$, can also be viewed as the regularized solution to the constrained optimization problem (Altman, 2021; Gattami et al., 2021) defined as follows:

$$\max_\pi J(\pi;p_\tau) = \sum_{t=0}^{T} \gamma^t \mathbb{E}_{a_t\sim\pi,s'\sim p_\tau}\left[r^e(s_t,a_t)\right] \quad \text{s.t.} \quad \sum_{t=0}^{T} \gamma^t \mathbb{E}_{a_t\sim\pi,s'\sim p_\tau}\left[r^i(s_t,a_t)\right] \geq \eta \;,$$

where $\eta \geq 0$ is some desired level of cumulative exploration. Intrinsic rewards, or exploration bonuses, have been defined in a variety of different ways in the literature. Some popular methods include, but are not limited to, intrinsic rewards based on:

i) *Prediction Error* (Schmidhuber, 1991; Stadie et al., 2015; Pathak et al., 2017), where $r_t^i$ is defined as $r_t^i = \frac{\eta}{2}\|f_\theta(s_t,a_t) - s_{t+1}\|_p$, $\|\cdot\|_p$ denotes the $L$ space norm and $f_\theta(\cdot,\cdot)$ is a predictive model for the environment dynamics, i.e. $\hat{\mathbb{E}}[p(s_{t+1}|s_t,a_t)] = f_\theta(s_t,a_t)$ (e.g., a neural network).

ii) *Variance* (Hester & Stone, 2012), where $r_t^i$ can be defined as $r_t^i = \mathbb{E}\big[\|f_\theta(s_t,a_t) - \mathbb{E}_\theta[f_\theta(s_t,a_t)]\|_2^2\big]$, and $f_\theta(\cdot,\cdot)$ again is a model of the environment dynamics.

iii) *Visitation Frequency* of state-action pairs $(s_t,a_t)$ (Bellemare et al., 2016; Tang et al., 2017; Ostrovski et al., 2017), where $r_t^i$ is defined as $r_t^i = \eta N(s_t,a_t)^{-1}$, and $N(s_t,a_t)$ is the count of encountered $(s,a)$ pairs. Pseudo-counts are typically used with high-dimensional $\mathcal{S} \times \mathcal{A}$.

iv) *Empowerment* (Klyubin et al., 2005; Salge et al., 2014), where $r_t^i = \eta\, I(s_{t+1}, a_t | s_t = s)$. Here, $I(\cdot, \cdot | \cdot)$ is the conditional mutual information quantifying the amount of control the agent can exert via $a_t$ in a specific state $s_t = s$. Empowerment based intrinsic rewards $r_t^i$ nudge the agents towards more controllable states.

A drawback of some intrinsic reward approaches, especially when transitions are noisy, is that they are not guaranteed to diminish as the agent becomes more familiar with the environment, since they conflate epistemic uncertainty with aleatoric uncertainty. This persistence can lead to excessive exploration of already well-understood state-action pairs, hindering the convergence to high reward regions (Chentanez et al., 2004; Burda et al., 2018). Furthermore, when intrinsic rewards are incorporated into the learning process, they effectively modify the original MDP by altering the reward function. This implies that the resulting value and discounted return functions $Q^{\pi, p_\tau}(s, a)$, $V^{\pi, p_\tau}(s)$ and $J(\pi; p_\tau)$ are effectively 'biased' with respect to the main task described by the original MDP. If the intrinsic reward modification is persistent, it may bias trained policies away from optimal solutions. Therefore, it is essential to design intrinsic rewards that facilitate efficient exploration at early stages of training, by disentangling epistemic and aleatoric uncertainties, and at the same time naturally fade to zero once the local environment dynamics is well understood.

## 2.2 Bayes-Adaptive MDPs and Exploration as a Complementary Task

A grounded way of representing the agent's complementary task of exploration, such that it is consistent with an MDP's state values functions, can be devised by adopting a Bayesian perspective on the uncertainty stemming from the environment dynamics. To this end, let us define the true parameters underlying the transition distribution of the dynamics as $\theta_{true} \in \Theta$, where we equivalently use $p(s_{t+1}, r_t | s_t, a_t; \theta_{true}) \equiv p_{\theta_{true}}(s_{t+1}, r_t | s_t, a_t)$. The typical Bayesian approach entails placing some form of prior distribution $p(\theta)$ on the unknown $\theta_{true}$, define a likelihood on the collected data $p(\mathcal{D}^n | \theta)$, where $n$ is the sample size, and update beliefs on it after collecting new data $\mathcal{D}^n$ through Bayes rule, $p(\theta | \mathcal{D}^n) \propto p(\mathcal{D}^n | \theta) p(\theta)$. With this in mind, we can formulate a modified version of the classic MDP, **Bayes-Adaptive Markov Decision Process** (BAMDP) (Duff, 2002; Ross et al., 2007), which explicitly incorporates belief updates over $\theta$ into the agent's decision-making process and enables one to define intrinsic rewards as a function of the agent's current belief over $\theta$:

**Definition 2.1** (BAMDP). A BAMDP is a tuple $\langle \mathcal{H}_\mathcal{S}, \mathcal{A}, p_h, p_0, r^h, \gamma \rangle$, defined over an horizon of $t \in \{1, ..., T\}$ time steps where:

i) $\mathcal{H}_\mathcal{S} = \mathcal{S} \times \Theta$ is the set of hyper-states, defined as the Cartesian product of the environment states space $\mathcal{S}$ and the space of parameters of the posterior transition dynamics $\Theta$;

ii) $\mathcal{A}$ is the action space;

iii) $p_h(\cdot | h_{s_t}, a_t) \in \mathcal{P}_\tau$ is the hyper-state transition probability that determines the next hyper-state $(s_{t+1}, \theta')$ given the current hyper-state $(s_t, \theta)$ and action $a_t$, and is such that $p_{h_t}(s_{t+1}, \theta' | s_t, a_t; \theta) = p(s_{t+1} | s_t, a_t; \theta) p(\theta' | s_t, a_t; \theta)$. The probability $p(\theta' | (s_t, a_t, s_{t+1}); \theta)$ is then the updated belief posterior $p(\theta' | \mathcal{D}; \theta)$ on $\theta_{true} \in \Theta$, given the revealed new state $s_{t+1}$ that completes the full transition $(s_t, a_t, s_{t+1}) \equiv \mathcal{D}$ (i.e., one new data point);

iv) $p_0 \in \mathcal{P}_0$ is the joint distribution of the initial state and the prior on the dynamics parameters, which can be assumed to be independent, i.e., $p_0(s_0, \theta) = p(s_0) p(\theta)$, where $s_0 \sim p(s_0)$;

v) $r^h : \mathcal{H}_\mathcal{S} \times \mathcal{A} \to \mathcal{C} \subset \mathbb{R}$ is a bounded combined reward function that stems from the hyper-state $\mathcal{H}_\mathcal{S} = \mathcal{S} \times \Theta$ and action $\mathcal{A}$ spaces and include both extrinsic rewards $r^e$, strictly generating from $\mathcal{S} \times \mathcal{A}$, and intrinsic rewards $r^i$, generating from $\Theta$, as discussed below;

vi) $\gamma \in (0, 1)$ is a discount factor.

Our definition of a BAMDP differs slightly from that typically found in the literature on Bayesian RL (Duff, 2002; Ross et al., 2007; Ghavamzadeh et al., 2015), as we added an explicit decomposition of the total rewards $r^h$ into extrinsic rewards $r^e$, stemming uniquely from $\mathcal{S} \times \mathcal{A}$, and intrinsic rewards $r^i$ stemming from $\Theta$, i.e., $r^i : \Theta \to \mathcal{C} \subset \mathbb{R}$. A BAMDP explicitly incorporates the agent's updated belief about the unknown dynamics parameter, expressed via the posterior distribution $p(\theta | \mathcal{D})$ — in the short form notation where $\mathcal{D} = (s, a, s')$ — into the decision-making process, in addition to expressing the complementary task of

exploring the environment to learn this parameter through the intrinsic rewards defined as a function of the current belief on $\theta$, $r^i(\theta)$. We have intentionally kept the definition of $r^i(\theta)$ vague for now, as in the next section we will discuss how to develop a measure for $r^i(\theta)$ with certain desirable properties, such that the BAMDP aligns back with the original underlying MDP asymptotically during training, through its value functions and discounted cumulative returns.

# 3 Information-Theoretic Intrinsic Rewards

In this section, we discuss a principled approach to define the intrinsic reward $r^i(\theta)$ using information-theoretic measures. Specifically, we focus on the concept of *Bayesian Surprise* (Itti & Baldi, 2009), which quantifies the amount of information acquired about the unknown dynamics parameters $\theta \in \Theta$ after observing new data $\mathcal{D}^n$ of sample size $n$. By incorporating an information-theoretic measure into the intrinsic reward, we encourage the agent to explore areas of the state-action space $\mathcal{S} \times \mathcal{A}$ that are most informative in terms of uncertainty reduction from a prior $p(\theta)$ to the posterior $p(\theta|\mathcal{D}^n)$.

## 3.1 Epistemic versus Aleatoric Uncertainty

The total variation residing in the random variable $\theta \in \Theta$ can be quantified with the Shannon entropy $H[p(\theta)] = \mathbb{E}_{\theta \sim p}[-\log p(\theta)]$ (Shannon, 1948). The total entropy $H[p(\theta)]$ can be decomposed, with respect to another random variable $Y$, as $H[p(\theta)] = H[p(\theta|Y)] + I(\theta; Y)$ (Cover, 1999). Here, the conditional entropy $H[p(\theta|Y)]$ quantifies the residual uncertainty after observing all the realization of $Y = y$, i.e., aleatoric uncertainty. The mutual information component, $I(\theta; Y)$, measuring the uncertainty reduction in one variable after observing the other, incorporates all the epistemic components. Conditioning everything on some $n$ sized dataset $\mathcal{D}^n$, we obtain:

$$I(\theta; Y|\mathcal{D}^n) = H[p(\theta|\mathcal{D}^n)] - H[p(\theta|Y, \mathcal{D}^n)] = \mathrm{IG}_\theta(Y, \mathcal{D}^n) \ , \tag{1}$$

where the conditional mutual information $I(\theta; Y|\mathcal{D}^n) = \mathrm{IG}_\theta(Y, \mathcal{D}^n)$ is typically referred to as **Information Gain** (Lindley, 1956) and it measures the reduction in the entropy of parameters $\theta \in \Theta$ before and after conditioning on the new random variable $Y$. Thus, intuitively $\mathrm{IG}_\theta(\cdot)$ captures the epistemic uncertainty gains (Hüllermeier & Waegeman, 2021; Wimmer et al., 2023) relative to parameters $\theta \in \Theta$, as we progressively gather more data.

Information gain $\mathrm{IG}_\theta(\cdot)$ appears to be an appealing candidate for a measure of intrinsic reward. In the context outlined by (BA)MDPs the information gain can be written in full form as:

$$\begin{aligned} \mathrm{IG}_\theta(s_t, a_t, s_{t+1}) &= H[p(\theta|s_t, a_t)] - H[p(\theta|s_t, a_t, s_{t+1})] \\ &= \mathbb{E}_{p(\theta|s_t, a_t, s_{t+1})}[\log p(\theta|s_t, a_t, s_{t+1})] - \mathbb{E}_{p(\theta|s_t, a_t)}[\log p(\theta|s_t, a_t)], \end{aligned} \tag{2}$$

where, similarly to before, $\mathrm{IG}_\theta(s_t, a_t, s_{t+1})$ measures the reduction in uncertainty around $\theta \in \Theta$ after observing the transition $(s_t, a_t, s_{t+1})$. Houthooft et al. (2016) were the first to leverage an information gain $\mathrm{IG}_\theta(\cdot)$ bonus such as the one described above, by specifically employing a variational Bayesian Neural Network to compute a one-step IG inside a model-free policy gradient learner. Their method neither embeds IG into a multi-step planner nor provides any formal analysis of bonus decay or convergence.

**Relation to Visitation-Frequency Bonuses** Bellemare et al. (2016) show that both exact counts and pseudo-counts can be interpreted as the Bayesian Information Gain of an agent that maintains a density model of the observations encountered; the corresponding pseudo-count (or count) exploration bonus can therefore be viewed as an explicit measure of epistemic uncertainty as well. Earlier count-based algorithms such as MBIE-EB (Strehl & Littman, 2008) and UCRL2 (Auer et al., 2008) already exploited this idea implicitly by adding a bonus $\beta/\sqrt{N(s, a)}$ to guarantee optimism and PAC-MDP or regret bounds. Subsequent density-model variants — e.g. hashing–pseudo-counts (Tang et al., 2017) and PixelCNN pseudo-counts (Ostrovski et al., 2017) — retain the same uncertainty interpretation while scaling to high-dimensional inputs. In these works the intrinsic reward decays at the non-asymptotic rate $O(1/\sqrt{N(s, a)})$. A key distinction with the dynamics model-based approach considered in this work however lies in the fact that forward dynamics

models naturally provide both next-state and reward predictions, together with the possibility of producing uncertainty measures for H-step-ahead predictions. In contrast, visitation-frequency bonuses depend on future counts, which are less straightforward to incorporate into look-ahead planning, as both a density model and a forward dynamics model would need to be specified.

## 3.2 Why Information Gain Intrinsic Rewards?

The intrinsic reward defined as $r_t^i(\theta) = \mathrm{IG}_\theta(s_t, a_t, s_{t+1})$ possesses the desirable property that it is guaranteed to gradually diminish as the agent explores the environment. This happens because, as the agent collects more transitions from the environment, its posterior belief over the dynamics parameters $\theta \in \Theta$ becomes increasingly concentrated — thus, the epistemic information gain from new observations naturally decreases. To formally demonstrate this, assume for simplicity $\mathcal{S}, \mathcal{A} \subseteq \mathbb{R}$ and that the transition distribution $p_{\theta_{true}}(s'|s, a)$ with true parameters $\theta_{true} \in \Theta$ follows a Markov dynamics characterized by the following additive noise structural equation:

$$S_{t+1} = f(S_t, A_t) + \varepsilon_t, \quad \text{where} \quad \varepsilon_t \sim \mathcal{N}(0, \sigma^2) \quad \text{and} \quad \sigma^2 \in \mathbb{R}^+ , \tag{3}$$

and where $f \in \mathcal{F}$ is some arbitrarily complex function. Suppose we want to model the two unknown parameters in (3), i.e., $\theta = (f, \sigma^2) \in \mathcal{F} \times \mathbb{R}^+$, in a Bayesian way, by placing a prior $p(\theta) = p(f, \sigma^2) = p(f)p(\sigma^2)$. Consider the $\mathrm{IG}_\theta(\cdot)$ associated with a specific one-step transition $(s, a, s')$, i.e., $\mathrm{IG}_\theta(s, a, s')$, so that the data $\mathcal{D}^n$ in this setting is equal to $\mathcal{D}^n \equiv \{(s_i, a_i, s_i')\}_{i=1}^n$. Under the mild assumptions of $\theta_{true} \in \mathrm{KL\text{-}support}(p(\theta))$ and "testability" (Schwartz, 1965; Ghosal et al., 1999) discussed in the Appendix, Section A.1, we have that:

**Proposition 3.1** (Consistency). *Assume the true data generating model in (3) with true parameters $\theta_{true} = (f_{true}, \sigma_{true}^2)$, a prior distribution $p(\theta) \in \mathcal{P}$ and the induced posterior $p(\theta \mid \{(s_i, a_i, s_i')\}_{i=1}^n)$. Given conditions for weak posterior consistency (discussed in the proof), such that for $\epsilon > 0$,*

$$p(\theta \in \Theta : d((f, \sigma), (f_{true}, \sigma_{true})) > \epsilon \mid \{(s_i, a_i, s_i')\}_{i=1}^n) \overset{P_{\theta_{true}}}{\to} 0 ,$$

*as $n \to \infty$, then $r_t^i(\theta) = IG_\theta(s, a, s') \overset{P_{\theta_{true}}}{\to} 0$ as $n \to \infty$.*

More practically, since we are interested primarily in carrying out Bayesian inference on the unknown function $f \in \mathcal{F}$ for exploration purposes, we might also treat $\sigma^2$ as fixed and estimate it via maximum likelihood, while instead maintaining only a Bayesian prior on $p(f)$ (e.g., Gaussian Process, Bayesian Neural Networks, Deep Ensembles, etc.).

Under the mild assumptions of Proposition 3.1, we can guarantee that the exploration incentive $r_t^i = \mathrm{IG}_\theta(\cdot)$ associated with a full transition $(s, a, s')$ asymptotically converges to zero, as the agent progressively learns $\theta \in \Theta$ via collecting new samples. Notice that this implies that the state-value (and action-value) function elicited by the augmented rewards $r_t = r_t^e + \eta_i r_t^i(\theta) = r_t^e + \eta_i \mathrm{IG}_\theta(\cdot)$, which characterize the BAMDP defined above in 2.1, convergences in probability to the original MDP's value functions. More formally, this reads:

**Corollary 3.2** (Value Function Convergence). *Given the state-value function associated with the BAMDP, $V_{BAMDP}^{\pi, p_\tau}(s)$, where $r_t^i(\theta) = IG_\theta(\cdot)$ defined as:*

$$V_{BAMDP}^{\pi, p_\tau}(s) = \mathbb{E}_{a \sim \pi(\cdot|s)}[r^e(s, a) + \eta_i IG_\theta(s, a, s') + \gamma \mathbb{E}_{p_\tau}[V_{BAMDP}^\pi(s')]] ,$$

*and the value function $V_{MDP}^{\pi, p_\tau}(s)$ associated with the original MDP, then we have that, under the conditions outlined in Proposition 3.1, $V_{BAMDP}^{\pi, p_\tau}(s) \overset{P}{\to} V_{MDP}^{\pi, p_\tau}(s)$ as $n \to \infty$. A similar result holds for the action-value function $Q_{BAMDP}^{\pi, p_\tau}(s, a) \overset{P}{\to} Q_{MDP}^{\pi, p_\tau}(s, a)$ and the discounted returns $J_{BAMDP}(\pi; p_\tau) \overset{P}{\to} J_{MDP}(\pi; p_\tau)$.*

These results ultimately establish, albeit asymptotically, the soundness of information gain as a desirable intrinsic reward signal, ensuring that it effectively drives exploration in the early stages of learning, while naturally diminishing as the agent becomes more familiar with the environment.

## 3.3 Information Gain Decay and Problem Complexity

While Proposition 3.1 guarantees that the intrinsic reward $r_t^i(\theta) = \mathrm{IG}_\theta(s_t, a_t, s_{t+1})$ asymptotically converges to zero as the number of observations increases, the rate at which this occurs is influenced by factors affecting

the *hardness* of the $f \in \mathcal{F}$ estimation problem, such as the signal-to-noise ratio and the smoothness of the true function $f_0$, and the type of prior $p(f)$ that is placed on $f$. For example, suppose we consider a Gaussian Process (GP) regression (Rasmussen et al., 2006) prior: $f|\omega \sim \mathcal{GP}\big(0, C(\cdot, \cdot|\omega)\big)$, where $C(\cdot, \cdot|\omega)$ is a kernel covariance function with hyper-parameters $\omega \in \Omega$. We can further derive minimax contraction rates $\epsilon_n$ (Ghosal & Van der Vaart, 2017) at which $r_t^i(\theta) = \mathrm{IG}_\theta(\cdot)$ converges by imposing functional form restrictions on the true underlying $f_{true} \in \mathcal{F}$. If we assume that: $\mathcal{X} = \mathcal{S} \times \mathcal{A} = [0,1]^{|\mathcal{A}|+1}$; $f_0 \in C^\alpha(\mathcal{X}) \cap H^\alpha(\mathcal{X})$, where $\mathcal{C}^\alpha(\cdot)$ is the Hölder space and $H^\alpha(\cdot)$ is the Sobolev space of order $\alpha$; $C(x,y) = \omega_1\|x-y\|^\alpha K_\alpha(\omega_2\|x-y\|)$ in $f|\boldsymbol{\omega} \sim \mathcal{GP}\big(0, C(\cdot, \cdot|\boldsymbol{\omega})\big)$ is the Matérn kernel, then we have (Van Der Vaart & Van Zanten, 2011):

**Proposition 3.3** (Contraction Rates). *Under the same assumptions of 3.1, if* $(f_0, \sigma_0) \in C^\alpha(\mathcal{X}) \cap H^\alpha(\mathcal{X}) \times [c,d]$, *where* $\mathcal{X} = [0,1]^{|\mathcal{A}|+1}$, *and* $f|\boldsymbol{\omega} \sim \mathcal{GP}\big(0, C(\cdot, \cdot|\boldsymbol{\omega})\big)$ *where* $C(\cdot, \cdot|\boldsymbol{\omega})$ *is the Matérn kernel, then:*

$$IG_\theta(s_t, a_t, s_{t+1}) \overset{P_0^\infty}{\to} 0 \quad as \quad n \to \infty \;,$$

*at the optimal minimax rate* $\epsilon_n = n^{-\frac{1}{(2+|\mathcal{X}|/\alpha)}}$ *(Yang & Barron, 1999).*

The optimal minimax rate $\epsilon_n$ achieved in Proposition 3.3 typically pertains to a purely 'passive' or random sampling strategy, and can potentially be improved under certain conditions through 'active' sampling (Willett et al., 2005; Castro & Nowak, 2008; Hanneke & Yang, 2015). We include a brief discussion on this in the Appendix, Section A.3. Also, note that $V_{\mathrm{BAMDP}}^{\pi, p_\tau}(s) \overset{P}{\to} V_{\mathrm{MDP}}^{\pi, p_\tau}(s)$ obviously occurs at the same rate.

Notice that the same type of asymptotic convergence characterizing $\mathrm{IG}_\theta(\cdot)$ is not guaranteed for other types of intrinsic rewards $r_t^i$, such as those based on prediction error $r_t^i = \frac{\eta}{2}\|f_\theta(s_t, a_t) - s_{t+1}\|_p$ (Stadie et al., 2015; Pathak et al., 2017) in stochastic environments. In settings such as the one described in the 'noisy TV problem' (Tang et al., 2017), prediction error can remain persistently high due to inherent *aleatoric uncertainty*, randomness in the environment that cannot be reduced through further data collection.

As a last note, we briefly comment on a natural point of comparison between the convergence results derived above and those concerning the visitation-based bonuses. While our contraction rate $\epsilon_n = n^{-1/(2+|\mathcal{X}|/\alpha)}$ pertains to the information gain (IG) bonus under smoothness assumptions on $f_0 \in \mathcal{F}$ and a GP prior, visitation-frequency-based bonuses typically rely on a different statistical setup. In the visitation-based framework, each $(s,a)$ pair is treated as an independent cell, and the exploration bonus decays as $O(1/\sqrt{N(s,a)})$ where $N(s,a)$ is the visit count. This typically assumes a tabular or "discretizable" domain, via pseudo-counts (Bellemare et al., 2016) — or at least that a good density model is learnable (Ostrovski et al., 2017) —, and does not explicitly model structure in the transition function. In contrast, our analysis assumes a continuous domain $\mathcal{X} = \mathcal{S} \times \mathcal{A}$, where the decay of IG is tied to the learnable complexity (e.g., smoothness $\alpha$) of the transition dynamics $f \in \mathcal{F}$. Thus, while the two forms of decay arise from different assumptions, our results can provide a principled guide in continuous domains where discretization or pseudo-counts are intractable or hard to interpret.

## 4 Efficient Model-Based Bayesian Exploration

While we have demonstrated above that defining intrinsic rewards via information gain $r_t^i(\theta) = \mathrm{IG}_\theta(s_t, a_t, s_{t+1})$ is theoretically appealing, this poses two practical challenges for its implementation in RL, and especially in MBRL domains. Firstly, computing $\mathrm{IG}_\theta(s_t, a_t, s_{t+1})$ for each timestep requires a posterior update over $\theta$ after observing each new transition $(s_t, a_t, r_t, s_{t+1})$. Maintaining and updating exact Bayesian posteriors in high-dimensional parameter spaces can be computationally expensive, especially when $\theta$ parameterizes complex neural networks, as in Houthooft et al. (2016). Secondly, full $\mathrm{IG}_\theta(\cdot)$ for a state-action pair $(s_t, a_t)$ is only calculated after the next state $s_{t+1}$ is observed and the posterior is updated accordingly, $p(\theta|s_t, a_t, s_{t+1})$. Hence, similarly to other intrinsic rewards, $\mathrm{IG}_\theta(s_t, a_t, s_{t+1})$ can only be computed "in retrospect", which makes it sample inefficient and complicates its integration into planning or policy optimization MBRL algorithms that instead typically look ahead.

A computationally more tractable, and planning-friendly, alternative is to employ the posterior predictive distribution directly, which marginalizes over the $\theta$ parameters, i.e., $p(s_{t+1}|s_t, a_t) = \int p(s_{t+1}|s_t, a_t; \theta) p(\theta|s_t, a_t) \, d\theta$, instead of the updated posterior distribution over the parameters after seeing $s_{t+1}$, i.e., $p(\theta|s_t, a_t, s_{t+1})$. However, in this scenario, we need to define a predictive surrogate for $\mathrm{IG}_\theta(s_t, a_t, s_{t+1})$, i.e., an Expected

Information Gain, $\text{EIG}_\theta(s_t, a_t)$, that marginalizes over the distribution of possible next states $s_{t+1}$. In this way, we can avoid explicitly updating the posterior for every new transition. The surrogate $\text{EIG}_\theta(s_t, a_t)$ can be derived as follows:

$$
\begin{aligned}
\text{EIG}_\theta(s_t, a_t) &= \mathbb{E}_{p(s_{t+1}|s_t, a_t)}\big[\text{IG}_\theta(s_t, a_t, s_{t+1})\big] && (4)\\
&= \mathbb{E}_{p(\theta|s_t, a_t)p(s_{t+1}|s_t, a_t;\theta)}\big[\log p(\theta|s_t, a_t, s_{t+1}) - \log p(\theta|s_t, a_t)\big]\\
&= \mathbb{E}_{p(\theta|s_t, a_t)p(s_{t+1}|s_t, a_t;\theta)}\big[\log p(s_{t+1}|s_t, a_t;\theta) - \log p(s_{t+1}|s_t, a_t)\big]\\
&= H\big[p(s_{t+1}|s_t, a_t)\big] - \mathbb{E}_{p(\theta|s_t, a_t)}\big[H[p(s_{t+1}|s_t, a_t;\theta)]\big] \;, && (5)
\end{aligned}
$$

where we use the fact that $p(\theta|s_t, a_t, s_{t+1}) \propto p(\theta|s_t, a_t)p(s_{t+1}|s_t, a_t, \theta)$ by Bayes rule and $p(s_{t+1}|s_t, a_t) = \int_\theta p(s_{t+1}|s_t, a_t, \theta)p(\theta|s_t, a_t)d\theta$ by marginalization. Full derivation is provided in the Appendix, Section B.1. One can intuitively view $\text{EIG}_\theta(s_t, a_t)$ as the *expected* reduction in uncertainty over $\theta$, if the agent takes action $a_t$ in state $s_t$. Notice also that equivalently we can express $\text{EIG}_\theta(s_t, a_t)$ as $\text{EIG}_\theta(s_t, a_t) = H\big[p(s_{t+1}|s_t, a_t)\big] - \mathbb{E}_{p(\theta|s_t, a_t)}\big[H[p(s_{t+1}|s_t, a_t;\theta]\big] = \mathbb{E}_{p(\theta|s_t, a_t)}[D_{KL}(p(s_{t+1}|s_t, a_t;\theta)\|p(s_{t+1}|s_t, a_t))]$. Since $\text{EIG}_\theta(s_t, a_t)$ does not explicitly require intermediate posterior computations, we do not necessarily need to maintain an updated posterior $p(\theta \mid s_t, a_t)$ throughout the training. Instead, for instance, we can resort to approximate methods, such as an ensemble of probabilistic deep dynamics models $\{f_{\theta_m}(s_t, a_t)\}_{m=1}^M$, that approximate the predictive posterior via model averaging (Lakshminarayanan et al., 2017; Wilson & Izmailov, 2020). One can then show that, under a deep ensemble model of the dynamics, $\text{EIG}_\theta(s_t, a_t)$ is theoretically equivalent to computing the disagreement among the ensemble's predictive distribution (Hanneke et al., 2014; Shyam et al., 2019), that is:

$$
\begin{aligned}
\text{EIG}_\theta(s_t, a_t) &= \mathbb{E}_{p(\theta|s_t, a_t)p(s_{t+1}|s_t, a_t;\theta)}\big[\log p(s_{t+1}|s_t, a_t;\theta) - \log p(s_{t+1}|s_t, a_t)\big]\\
&= \mathbb{E}_{p(\theta|s_t, a_t)p(s_{t+1}|s_t, a_t;\theta)}\Big[\log p(s_{t+1}|s_t, a_t;\theta) - \log \int_\Theta p(s_{t+1}|s_t, a_t;\theta)p(\theta|s_t, a_t)\, d\theta\Big]\\
&= H\big[\mathbb{E}_{p(\theta|s_t, a_t)}[p(s_{t+1}|s_t, a_t;\theta)]\big] - \mathbb{E}_{p(\theta|s_t, a_t)}\big[H[p(s_{t+1}|s_t, a_t;\theta)]\big]\\
&= \text{JSD}_\theta\big(p(s_{t+1}|s_t, a_t;\theta) \mid \theta \sim p(\theta|s_t, a_t)\big) \;, && (6)
\end{aligned}
$$

where JSD is the Jensen-Shannon Divergence. The intuition behind the equivalence in (6) is that, under a deep ensemble dynamics model, EIG can be computed as the level of disagreement across different plausible dynamics models consistent with the current data. Full derivation is provided in the Appendix, Section B.3. The above quantity can be approximated via Monte Carlo estimates by averaging the predictions of the probabilistic neural networks in the ensemble, as

$$
\begin{aligned}
\text{EIG}_\theta(s_t, a_t) &= H\big[\mathbb{E}_{p(\theta|s_t, a_t)}[p(s_{t+1}|s_t, a_t;\theta)]\big] - \mathbb{E}_{p(\theta|s_t, a_t)}\big[H[p(s_{t+1}|s_t, a_t;\theta)]\big]\\
&\approx H\left[\frac{1}{M}\sum_{m=1}^M p(s_{t+1}|s_t, a_t;\theta_m)\right] - \frac{1}{M}\sum_{m=1}^M H\big[p(s_{t+1}|s_t, a_t;\theta_m)\big] \;.
\end{aligned}
$$

Ensembles of probabilistic neural networks are a cheaper, approximate alternative to other Bayesian regression models such as Gaussian Processes or Bayesian Neural Networks, and work reasonably well when exact computation of the posterior (and posterior predictive) becomes intractable in high-dimensional state-action space settings.

## 4.1 Disagreement and Heteroskedastic Noise

An important distinction arises based on whether the environment's noise is homoskedastic or heteroskedastic. In the homoskedastic case, the transition noise variance $\sigma^2$ remains (theoretically) constant across all states, $\sigma(s) = \sigma$ for all $s \in \mathcal{S}$. This constancy means that *relative* differences in a predictive uncertainty measure (e.g., predictive entropy or variance) primarily reflect the model's epistemic uncertainty: if all states are subject to the same inherent noise, then the model's uncertainty should vary mainly due to gaps in its knowledge about the dynamics, rather than state-specific stochasticity. In practice, of course, approximation errors can still inflate or reduce these estimates, so the absolute values of such measures will not perfectly align with epistemic uncertainty — thus, using $\text{EIG}_\theta(\cdot)$ is still more efficient (see experiment in Section 5.1.1 below). However, their variations across states can still somehow provide a more meaningful exploration signal in a

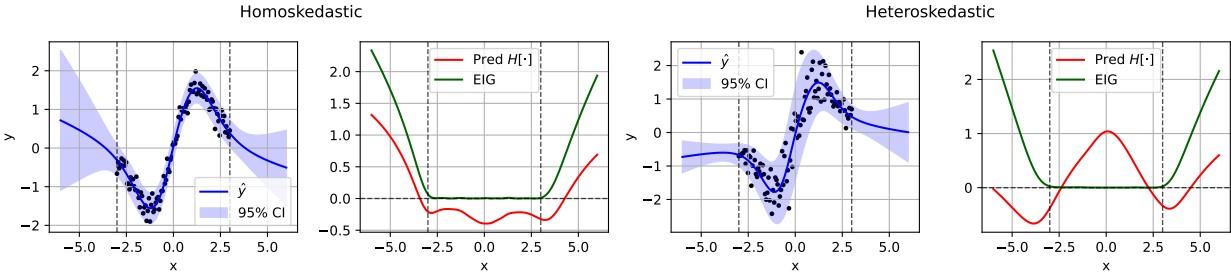

Figure 1: Left panels show a homoskedastic regression case, where predictive $H[\cdot]$ and $\text{EIG}_\theta(\cdot)$ identify similar regions of uncertainty. Right panels show a heteroskedastic regression case, where predictive $H[\cdot]$ is influenced by state-dependent noise, while $\text{EIG}_\theta(\cdot)$ still identifies regions of epistemic uncertainty. The regression model employed is an ensemble of probabilistic neural networks.

homoskedastic setting. In contrast, for heteroskedastic noise, where the noise variance $\sigma^2(s)$ depends on the state realization $s$, the same measures blend irreducible aleatoric uncertainty with epistemic uncertainty. As a result, highly noisy state regions may trigger misleading signals as they exhibit large predictive entropy (or variance) even if the model is confident about the underlying dynamics, simply because the environment itself is noisier.

Figure 1 illustrates these behaviors in a simple regression setting with either homoskedastic or heteroskedastic noise. Under homoskedastic noise, the relative levels of predictive entropy (in red) and information-theoretic exploration measure $\text{EIG}_\theta(\cdot)$ (in green) align closely with regions of true knowledge gaps. In contrast, for heteroskedastic noise, predictive entropy spikes in the (middle) region with higher noise, whereas $\text{EIG}_\theta(\cdot)$ remains low there because the model has ample data – i.e., $\text{EIG}_\theta(\cdot)$ explicitly factors out irreducible noise, focusing on states that would *reduce* the model's parameter uncertainty. This illustrates why we not only need an uncertainty-aware dynamics model but also need to carefully disentangle aleatoric and epistemic uncertainties when deriving exploration signals.

## 4.2 Predictive Trajectories Sampling with EIG Exploration Bonus

For some of the setups in the experimental Section 5 below, we will be using standard model-free policy gradient algorithms, namely Proximal Policy Optimization (PPO) (Schulman et al., 2017) for environments with discrete action spaces and Soft Actor-Critic (SAC) (Haarnoja et al., 2018) for ones with continuous action spaces. We generally employ vanilla versions of these algorithms, and versions that are augmented with intrinsic curiosity, i.e., the collected extrinsic rewards $r_t^e$ in the replay buffer $\mathcal{D} = \{s_t, a_t, r_t, s_{t+1}\}_{t=0}^T$ are augmented with intrinsic rewards $r_t^i$ computed using the forward dynamics model according to the description in Section 2.1.

For some other specific setups instead, we instead employ a general Predictive Trajectory Sampling with Bayesian Exploration (PTS-BE) framework, which extends prior planning-to-explore and model predictive control approaches (Chua et al., 2018; Shyam et al., 2019; Sekar et al., 2020) by supporting arbitrary Bayesian models of both dynamics and rewards. In this framework we maintain an ensemble of $M$ probabilistic models $\{f_{\theta_m}\}_{m=1}^M$, each predicting the joint distribution $p_{\theta_m}(r, s' \mid s, a)$. Notice that here we generally consider prediction of both reward and next state as part of an encompassing dynamics prediction effort, but one can easily focus only on next state prediction, particularly in settings where reward prediction is secondary. Because $r$ and $s'$ are conditionally independent given $(s, a)$, we factorize each $p_{\theta_m}(r, s' \mid s, a) = p_{\theta_m}(r \mid s, a)\, p_{\theta_m}(s' \mid s, a)$. Under the usual assumption that $(r, s')$ form a multivariate Gaussian with diagonal covariance, deep ensembles provide a scalable, approximate Bayesian predictive posterior without explicit posterior updates. Instead of selecting actions through optimization procedures like Random Shooting or CEM, as in traditional Model Predictive Control (Draeger et al., 1995; Chua et al., 2018), we instead maintain a policy parametrization $\pi_\psi(s)$ (e.g., PPO, SAC, etc.) that does not necessarily require updates at each time step $t$, as proposed also in Shyam et al. (2019) and Foster et al. (2021). Similarly to model-free algorithms, this policy is trained by

---

**Algorithm 1** Predictive Trajectory Sampling with Bayesian Exploration (PTS-BE)

---
1: Initialize policy $\pi_\psi$, transitions buffer $\mathcal{D}_{env}$, Bayesian dynamics model $p(f)$ (e.g., GP, Deep Ensembles, ...)
2: **for** $T$ steps **do**
3:     Given state $s_t$, sample action $a_t \sim \pi_\psi(s_t)$; receive $r_t$ and $s_{t+1}$
4:     Add $\mathcal{D}_{env} \leftarrow \mathcal{D}_{env} \cup \{s_t, a_t, r_t, s_{t+1}\}$
5:     **if** $t > t_{warm}$ **then**
6:         **if** $t \bmod t_{policy} = 0$ **then**
7:             Sample $K$ actions from policy $\{a_t^{(k)}\}_{k=1}^K \sim \pi_\psi(s_t)$
8:             Rollout $J$-length trajectories and define $\mathcal{D}_{model} = \{\{(s_{t+j}^{(k)}, a_{t+j}^{(k)}, r_{t+j}^{(k)}, s_{t+1+j}^{(k)})\}_{j=0}^J\}_{k=1}^K$
9:             Compute $r_{t+j}^i$ and augment each $r_{t+j}^{(k)}$ in $\mathcal{D}_{env}$ as $r_{t+j}^{(k)} = r_{t+j}^e + \eta_t r_{t+j}^i = r_{t+j}^e + \eta_t \text{EIG}_\theta(s_{t+j}^{(k)}, a_{t+j}^{(k)})$
10:             **for** $G$ policy gradient updates **do**
11:                 Update $\psi$ in $\pi_\psi(s)$ using $\mathcal{D}_{model}$
12:         **if** $t \bmod t_{model} = 0$ **then**
13:             **for** $M$ model updates **do**
14:             Update dynamics model $p(f|D_{env}) \leftarrow p\big(f|D_{env}, (s_t, a_t, r_t, s_{t+1})\big)$

---

maximizing the cumulative sum of both extrinsic rewards $r^e$ and intrinsic rewards $r^i$, $\sum_{t=1}^T (r_t^e + \eta r_t^i)$, where $r^i$ again can be generally defined as an output of the predictive model for $p_\theta(r, s'|s, a)$ (e.g., Variance, Entropy, EIG, etc.), but uses a different data buffer. Specifically, the ensemble models are leveraged to generate $K$ independent predictive trajectories from the current state $s_t$, each unrolled for $J$ steps into the future, $\{(s_{t+j}^k, a_{t+j}^k, r_{t+j}^k, s_{t+j+1}'^k)\}_{j=1}^J$. Actions are drawn from the current policy $a_{t+j}^k \sim \pi_\psi(s_{t+j}^k)$, while next state and rewards are sampled from the predictive posterior $p_\theta(r, s'|s, a)$. Intrinsic rewards $r_t^i$ are computed for every $k$ and $j$ entry and added up to the estimated extrinsic ones, if included. Finally, the policy parameters $\psi$ are updated periodically every $n$ steps using purely model-generated data from the PTS step. One could also potentially consider using a combination of real and imagined trajectories, like in Model-Based Policy Optimization (Janner et al., 2019). A pseudo-code representation of the full algorithm's structure is depicted in Algorithm 1.

**Obtaining $p(f|D)$ for different forward models.** Line 14 in Algorithm 1 yields an updated belief posterior $p(f|D_t)$ on the environment dynamics. In practice, the way this belief posterior is obtained depends on which Bayesian dynamics model is used. In particular: (i) Exact GPs maintain a full multivariate Gaussian posterior $p(f \mid D)$, using $(s, a)$ as inputs; ii) DKs maintain a multivariate Gaussian posterior using latent input representations $\phi(s, a)$, where the marginalization is identical to the full GP, but performed in a latent space $\mathcal{H}$; iii) Stochastic Variational variants of GPs and DKs (SVGP / SVDK) maintain a Gaussian posterior using a subset of representative inducing points in either $(s, a)$ or $\phi(s, a)$, respectively — the marginalization is again identical to the GP or DK, but carried out using only the inducing points; iv) Deep Ensembles (DEs) approximate the full belief posterior via an empirical mixture of $M$ probabilistic neural networks — marginalization is obtained via Monte Carlo averaging: $\hat{p}(f \mid D) = \frac{1}{M} \sum_{m=1}^M \delta_{f_{\theta_m}}$.

## 5 Experiments

In this section, we evaluate the performance and the practical benefits of information-theoretic exploration bonuses through three sets of experiments. The first set is designed to validate the theoretical claims made in earlier sections and demonstrate the advantages of the proposed PTS-BE (Algorithm 1) in terms of planning deep exploration strategies. The second set introduces a simple unichain environment to allow comparison of the approximate uncertainty quantification properties of deep ensembles with those of Gaussian Processes (Rasmussen et al., 2006), which provide full Bayesian predictive posterior, in terms of their ability to generate a reliable exploration signal based on epistemic uncertainty. In this second set of experiments, we include also Deep Kernels (Wilson et al., 2016b) in the comparison, and other popular curiosity-based algorithms that use intrinsic reward augmentation on the environment's data buffer to update the policy via exploration bonuses computed "in retrospect" (see discussion in Section 4). Deep Kernels combine the flexibility of deep learning with the principled uncertainty estimation of Gaussian Processes, making them a valuable baseline for assessing the trade-offs between expressiveness and calibrated uncertainty in exploration-driven tasks.

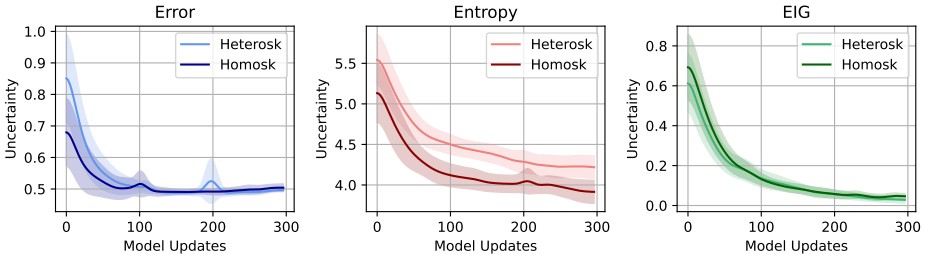

Figure 2: Intrinsic reward uncertainty measures on the homoskedastic and heteroskedastic versions of the Mountain Car environment, recorded at every update of the dynamics model $f_\theta(s_t, a_t)$, under a random policy. From left to right plot: prediction error $r_t^i = \|f_\theta(s_t, a_t) - s_{t+1}\|_2$, entropy $r_t^i = H[p_\theta(r, s'|s, a)]$ and expected information gain $r_t^i = \text{EIG}_\theta(\cdot)$.

Lastly, the third set of experiments showcases the applicability of the PTS-BE approach to more challenging, sparse-reward, Maze environments characterizing close to purely explorative tasks.

## 5.1 Heteroskedastic Noise Model in Mountain Car

We begin by introducing a modified version of the canonical Mountain Car environment. In our variant, we incorporate a heteroskedastic noise model based on the environment's physical dynamics. The transition dynamics are modified so that the noise variance is state-dependent, $S_{t+1} = f(S_t, A_t) + \varepsilon_t$ where $\varepsilon_t \sim \mathcal{N}(0, \sigma^2(S_t))$, thereby mimicking realistic scenarios where uncertainty varies with the agent's position and velocity. In particular, noise $\sigma^2(S_t)$ is higher for higher velocities (more chaotic dynamics at higher speeds) and for positions near the bottom of the valley (simulating factors such as different terrain conditions). Conversely, we define the homoskedastic version of the environment as one where the transition noise remains constant across all states, i.e., $\varepsilon_t \sim \mathcal{N}(0, \sigma^2)$. Throughout the experiments in this section, we make use of a deep ensemble to estimate the dynamics $\{f_{\theta_m}(s_t, a_t)\}_{m=1}^M$.

The first set of results reported in Figure 2 illustrates the values of different model-based uncertainty measures that can be used to define the intrinsic rewards $r_t^i$, in both the homoskedastic and heteroskedastic versions of Noisy Mountain Car. The measures are: i) mean-squared prediction error, defined as $r_t^i = \|f_\theta(s_t, a_t) - s_{t+1}\|_2$ (Stadie et al., 2015; Pathak et al., 2017); ii) entropy of the predictive distribution, defined as $r_t^i = H[p_\theta(r, s'|s, a)]$; iii) EIG, defined according to Equation 6, as we are specifically making use of a deep ensemble dynamics model in this context. Each uncertainty measure is computed at every model update, that is, every $t_{model} = 64$ environment steps, and transitions are gathered under a random policy. As can be seen from the plots of Figure 2, EIG progressively converges to zero as the number of model updates increase, while prediction error and entropy plateau at higher values. Moreover, EIG shows very similar progression both in the homoskedastic and heteroskedastic noise scenarios. This is because EIG, in the form of Jensen-Shannon Divergence here, is low in regions with high noise but rich in data, since even if the predictive variance in each ensemble member is high, the overall "disagreement" among the ensemble's predictive distributions, $\{p(r, s'|s, a; \theta_m)\}_{m=1}^M$, is low. Predictive entropy's values instead are consistently higher in the heteroskedastic case relative to the homoskedastic one, since aleatoric and epistemic uncertainties are fused in together in the measure, resulting in a stronger, but deceiving, exploration signal to the agent, as the relative difference is driven solely by aleatoric uncertainty. Finally, prediction error curves slightly diverge for approximately the first 50 model updates, then plateau at similar values. Notice that importantly, even though the prediction error curves do converge to similar values in this specific example, where we employ deep ensemble, the process of averaging predictions in the ensemble can mask substantial disagreement among ensemble members — meaning that even when the ensemble's mean prediction has low MSE, the underlying epistemic uncertainty may still be high. Similar reasoning naturally holds also in this case of non-ensemble, and more generally non-Bayesian, predictive models (e.g., a single neural net).

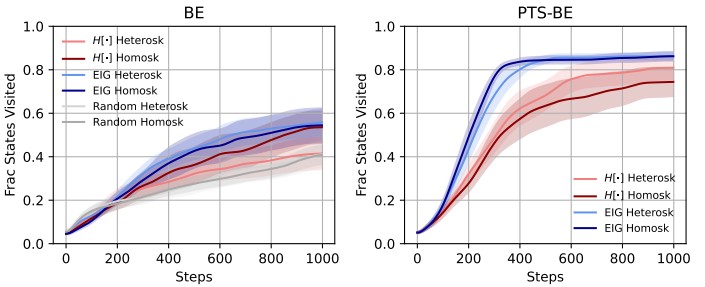

Figure 3: Results from the second experiment on Noisy Mountain Car. The two plots report the cumulative fraction of visited states by the agent over 1000 steps, while the tables report the average returns. These are computed over 20 different replications. The methods being compared are: a random agent (Random), a baseline model-free agent with intrinsic rewards augmentation (BE) and a model-based Predictive Trajectory Sampling (PTS-BE) agent, as described in Section 4.2. The BE and PTS-BE methods are compared on both the homoskedastic and heteroskedastic versions of the environment, using either predictive entropy $H[\cdot]$ or EIG as intrinsic reward. The baseline solver is PPO.

### 5.1.1 Planning to explore with PTS-BE

In the second set of results regarding the Noisy Mountain Car environment, we aim to assess the performance of EIG based intrinsic rewards (i.e., which we refer to as "Bayesian Exploration", or BE onward, for simplicity) in conjunction with Predictive Trajectory Sampling, in the form of the algorithm introduced in 1. To this end, we compare two types of algorithms. The first one is a baseline, model-free, PPO agent (Schulman et al., 2017) that uses intrinsic rewards augmentation and updates the policy retrospectively based on these augmented rewards in the buffer (we will denote this with the "BE" prefix). In short, at each policy update we form a dataset $D_T = \{(s_i, a_i, r_i^e, s_i')\}_{t=1}^T$ of environment interactions, then replace each reward $r_t^e$ with $r_t = r_t^e + \eta r_t^i$, where $r_t^i$ is the intrinsic bonus for that transition (Entropy or EIG) and $\eta$ its scaling factor. Then, we performance the usual PPO update. The second is the PTS-BE algorithm presented in Section 4.2, which instead leverages the learnt dynamics model to do Predictive Trajectory Sampling, and updates a policy using these sampled trajectories. We also include a random policy agent as an additional baseline ("Random"). The results of this experiment are reported in Figure 3. Both BE and PTS-BE are run with different configurations, that is: i) using either predictive entropy $H[\cdot]$ or EIG as intrinsic rewards (hence the different suffix); ii) under either the homoskedastic or heteroskedastic version of the Noisy Mountain Car environment. For the PTS-BE specifications, we use an horizon of $J = 100$ steps ahead and $K = 10$ independent rollouts. The results demonstrate the natural data efficiency of the PTS-BE method in planning for deep exploration, while BE methods would necessitate a lot more samples to achieve similar state space coverage. On the difference between intrinsic reward measures employed, we notice that EIG generally does better than predictive entropy both in the homoskedastic and heteroskedastic scenarios. Furthermore, we note that a random policy achieves under 40% state-space coverage in 1.000 steps, under 70% in 5.000 steps, and about 90% only after 10.000 steps. And in none of these instances does it solve the environment's task. By contrast, PTS-BE-EIG solves the task 18/20 times in the heteroskedastic setting and 17/20 in the homoskedastic one, in fewer than 1.000 steps. PTS-BE-$H[\cdot]$ solves it only 8/20 and 10/20 times, respectively, within the same budget, demonstrating once again the superiority of EIG as an intrinsic reward signal. Finally, we also emphasize how PTS-BE-$H[\cdot]$ exhibits higher variance than PTS-BE-EIG, as its error bands are significantly wider.

### 5.2 Dynamics Models Comparison

In the second set of experiments we introduce an alternative simple unichain environment, similar to the ones used by Osband et al. (2016) and Shyam et al. (2019). The aim of this last setup is to compare the performance of PTS-BE with different dynamics model specifications, in particular with Gaussian Processes (GPs) (Rasmussen et al., 2006) and Deep Kernels (DKs) (Wilson et al., 2016b). GPs are considered to

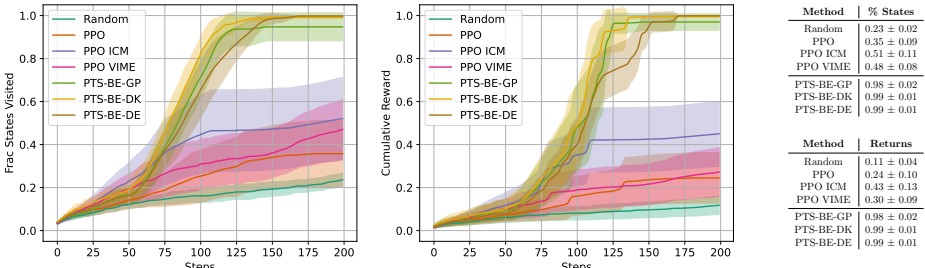

Figure 4: Results from the $L = 50$ states Unichain environment. The first plot reports cumulative fraction of visited states. The second plot reports cumulative rewards. The tables include fraction of visited states and cumulative rewards respectively, at episode termination. The models being compared are: a Random policy, PPO, PPO with ICM intrinsic rewards, PPO with VIME, and three version of the PTS-BE algorithm with different dynamics model specifications, i.e., GP, Deep Kernels (DK) and Deep Ensembles (DE).

be a gold standard for uncertainty quantification in regression problems, as they elicit a full, closed form, posterior predictive distribution on the output. The downside with GPs is that they notoriously scale poorly with sample size and input dimensions. In particular, in the context of dynamics MBRL modeling, their computational cost amounts to $\mathcal{O}(n^2|\mathcal{S} \times \mathcal{A}|)$ for evaluating the kernel matrix, in addition to the $\mathcal{O}(n^3)$ and $\mathcal{O}(n^2)$ costs incurred for matrix inversion at train and test time, respectively. Additionally, in MBRL, we typically model $|\mathcal{S}| + 1$ (states and rewards) outputs, which implies that we would need to maintain and update $|\mathcal{S}| + 1$ different GP models, increasing the cost by a factor proportional to $|\mathcal{S}| + 1$. For these reasons, deep ensembles are often preferred for their better scalability with input and output dimensions, albeit at the cost of having an approximate posterior predictive distribution. DKs (Wilson et al., 2016b) instead represent an alternative, principled way of reducing the input's computational cost in GPs. They first map inputs to a lower-dimensional latent space $\mathcal{H}$, $f_h : \mathcal{S} \times \mathcal{A} \to \mathcal{H}$, where $f_h$ is a neural net. Then a GP prior is placed on the latent space, $f_s : \mathcal{H} \to \mathcal{S}$. The parameters in both $f_h$ and $f_s$ are optimized end-to-end by minimizing a single loss. DKs reduce the kernel matrix evaluation costs from $\mathcal{O}(n^2|\mathcal{S} \times \mathcal{A}|)$ to $\mathcal{O}(n^2|\mathcal{H}|)$, where $|\mathcal{H}|$ is controllable by the user. However, they still bear the cost of maintaining $|\mathcal{S}| + 1$ GP models, one per output, on the latent space. Finally, we specify that in order to reduce the cost associated with sample size $n$ only, we employ sparse input variational methods for both GP and DK specifications (Titsias, 2009; Wilson et al., 2016a). This reduces the kernel evaluation cost from $\mathcal{O}(n^2|\mathcal{S} \times \mathcal{A}|)$ to $\mathcal{O}(nm|\mathcal{S} \times \mathcal{A}|)$, where $m$ is the size of the chosen sub-sample, and the matrix inversion cost from $\mathcal{O}(n^3)$ to $\mathcal{O}(m^3)$ at train time and from $\mathcal{O}(n^2)$ to $\mathcal{O}(m^2)$ at test time for GPs (for DKs, replace $|\mathcal{S} \times \mathcal{A}|$ with $|\mathcal{H}|$).

The unichain environment version we consider is made of $L = 50$ discrete states and three discrete actions: `{go-left, stay, go-right}`. We use a continuous representation of the discrete states as in Osband et al. (2016) and Shyam et al. (2019). The agent is spawned at the second state $s_0 = 2$. Rewards are sparse, equal to 0.001 in the first state and to 1.0 in the last state, while they are zero everywhere else. A visual depiction of the unichain environment is included in the Appendix, Figure 8, together with an additional experiment on a $L = 100$ states, noisy unichain environment. Results from the experiment are depicted in Figure 4. The plots measure the cumulative fraction of visited states and the cumulative rewards respectively, for each method. The tables instead report the fraction of visited states and the cumulative rewards achieved at episode termination. The list of methods compared is the following: a random agent (Random); a PPO agent (PPO); a PPO agent augmented via Intrinsic Curiosity Module (ICM) (Pathak et al., 2017), a popular method based on prediction error intrinsic rewards (PPO ICM); a PPO agent augmented with VIME intrinsic rewards (Houthooft et al., 2016), which are based on the IG specification introduced in Equation 2 (PPO VIME); lastly, we include three versions of the PTS-BE algorithm, using EIG as an intrinsic reward, with different dynamics model specifications based on GPs (PTS-BE-GP), Deep Kernels (PTS-BE-DK) and Deep Ensemble (PTS-BE-DE). The results again show superiority of the PTS approaches compared to popular methods such as ICM and VIME, for settings where deep look-ahead exploration is required to preserve sample efficiency. In addition, we note that GPs and DKs specification display only marginally better performance, as they elicit a full posterior predictive distribution, compared to Deep Ensembles. This demonstrate that, even if the

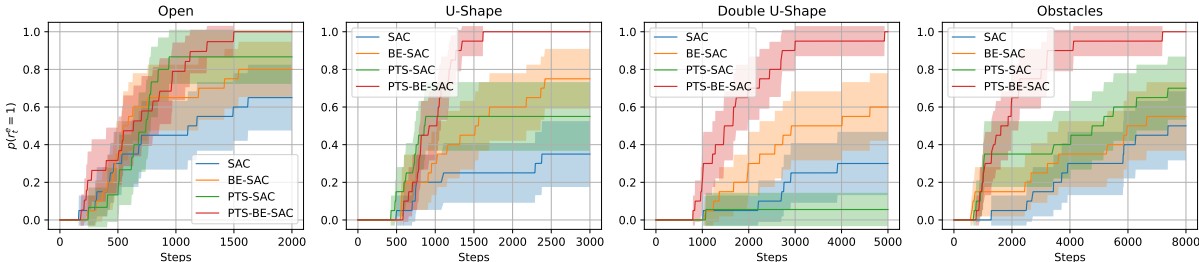

Figure 5: Results on the four different configurations of the maze environment, namely: i) an open maze with no obstacles ("Open"); ii) a U-shaped maze ("U Shape"; iii) a maze featuring two U turns ("Double U Shape"); iv) a large maze with several obstacles in the middle ("Obstacles"). The plots show the cumulative probability of reaching the goal at every step $t$, i.e., $p(r_t^e = 1)$, for each of the four methods compared: SAC, BE-SAC, PTS-SAC and PTS-BE-SAC.

posterior predictive distribution in Deep Ensemble is only approximate and may not have the same coverage properties of the full one in GPs, it still provides a sufficiently reliable signal for exploration.

### 5.3 Maze Exploration Environments

**Point Maze Environments**   This set of experiments features four types of 2D Gymnasium Robotics Point Maze structures (Fu et al., 2020) of increasing complexity. The agent's task in this setting is to explore the environment and collect a goal object located somewhere in the maze. The environments are characterized by continuous states and actions. Rewards are sparse, equal to zero everywhere except from in the goal state, where $r_t^e = 1$. The four environments include: i) an open maze with no obstacles ("Open"); ii) a U-shaped maze ("U-Shape"); iii) a medium size maze with double U-shape ("Double U-Shape"); iv) a larger maze with random obstacles in the middle ("Obstacles". The methods we compare here are: Soft Actor-Critc (SAC) (Haarnoja et al., 2018); SAC with Bayesian Exploration intrinsic reward augmentation (BE-SAC) based on EIG; a version of PTS planning algorithm without any intrinsic rewards, based only on extrinsic signal, using SAC as solver (PTS-SAC); lastly, the PTS-BE algorithm using SAC policy (PTS-BE-SAC). For the PTS-BE specification, we use an horizon of $J = 64$ and $K = 16$ independent rollouts. Results are reported in Figure 5. The plots show the cumulative probability of reaching the goal and solving the maze at every time step $t$, i.e. $p(r_t^e = 1)$, for each different maze configuration.

The results demonstrate that, even in the case of continuous control problems, PTS-BE guarantees better performance and sample efficiency, as it generally achieves the goal in significantly less steps. In particular, the gap in performance with respect to SAC, BE-SAC and PTS-SAC agents is smaller in the case of the first "Open" maze, as this is a considerably easier scenario to solve relative to the other configurations, while it becomes significantly larger in the other three setups where the task gets increasingly harder. The baseline PTS-SAC was included to further disentangle the contribution of model-based planning versus that of information gain intrinsic rewards. As expected, PTS-SAC alone does not really push performance up significantly, which confirms that the substantial gain particularly stems from the synergy between the information gain intrinsic rewards and the "planning-to-explore" approach, beyond planning alone.

**Ant Maze Environments**   The final set of experiments involves two different maze environments from the Gymnasium Robotics suite, featuring higher-dimensional state-action spaces. As with the Point Maze tasks, these Ant Maze environments were originally introduced in Fu et al. (2020). In these tasks, a robotic ant must learn to 'walk' from a starting location to a designated goal. The action space $\mathcal{A}$ consists of 8 dimensions corresponding to torques applied to the hinge joints, while the state space includes 27 positional features from various body parts. Although rewards are sparse by default (1 when the goal is reached, 0 otherwise), we incorporate a dense extrinsic reward signal to encourage locomotion behavior during training.

We evaluate performance in two maze variants — a small and a large open maze — using the same methods as in the Point Maze experiments: SAC, BE-SAC, PTS-SAC, and PTS-BE-SAC. Results are presented in

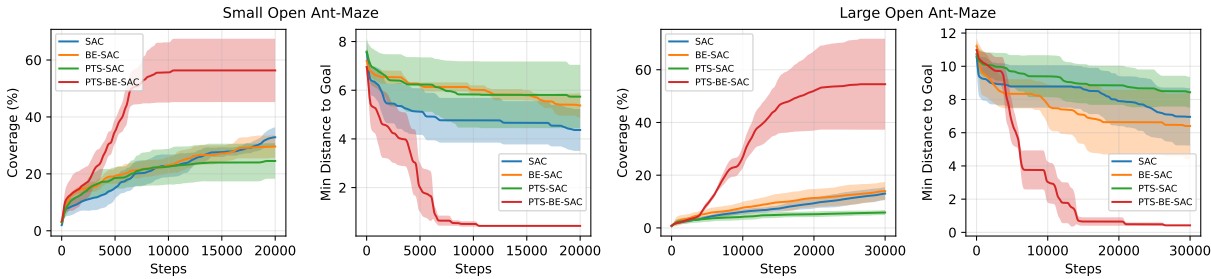

Figure 6: Results for the two Ant Maze environments, the Small and the Large Open mazes. For both environments we report the percentage of covered states ("Coverage (%)") and the minimum Euclidean distance to the goal state ("Min Distance to Goal") over 10 different runs. The task is solved when this distance is $\leq 0.5$. Methods compared are again SAC, BE-SAC, PTS-SAC and PTS-BE-SAC.

Figure 6. We report both the percentage of states covered ("Coverage (%)") and the minimum Euclidean distance to the goal achieved ("Min Distance to Goal") to assess not only exploration but also significant progress toward the target. A goal is considered reached when the agent's position is within a Euclidean distance of 0.5 from the goal coordinates. Consistent with prior results, PTS-BE-SAC is the only method among those compared that successfully solves the task in both environments, reaching the goal in an average of $7,281 \pm 1,531$ steps in the Small Open Maze and $15,237 \pm 4,372$ steps in the Large Open Maze.

## 6 Conclusions

In this work, we tackled the challenge of data-efficient exploration in MBRL by studying a principled information-theoretic approach to intrinsic motivation. Firstly, we derived a class of information-theoretic exploration bonuses grounded in Bayesian principles, specifically designed to target the epistemic uncertainty rather than the aleatoric uncertainty stemming from the environment ("Bayesian Exploration"). We then demonstrated, both theoretically and empirically, that, unlike other existing intrinsic rewards methods, these information gain bonuses have the desirable properties of naturally signaling epistemic gains and converging to zero asymptotically. We also discussed tractable equivalent measures that can be efficiently approximated with less computational effort via a deep ensembles dynamics model (Shyam et al., 2019). Finally, we introduced a general Predictive Trajectory Sampling with Bayesian Exploration (PTS-BE) framework — compatible with any Bayesian dynamics and reward model — that embeds these bonuses into multi-step planning. Empirically, PTS-BE outperforms reactive intrinsic-reward and model-free baselines on sparse-reward and pure-exploration tasks. As a final note, we conclude by stressing that, as with most planning-based MBRL methods, PTS-BE trades higher computational cost (maintaining a policy, a Bayesian model, and imagined rollouts) for improved sample efficiency. This trade-off is advantageous when environment interactions are expensive, while simpler intrinsic motivation methods may suffice when data is cheap.

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

# A    Proofs of Propositions

In order to prove the convergence properties of $\mathrm{IG}_\theta(\cdot)$ presented in the main body's propositions, we need to define a couple of necessary building blocks concepts (Ghosal & Van der Vaart, 2017). For this section, let us rename the prior distribution as $\pi(\theta)$, or shortly $\pi$, and the posterior as $\pi\big(\theta|(\xi_t)^n\big)$ for simplicity, where $\xi_t^n$ is a sequence of samples $n \in \{1, 2, ...\}$ following likelihood distribution $\xi^n \sim p(\xi^n|\theta)$ with $\theta \in \Theta$ and $\theta \in \pi(\theta)$. The posterior distribution associated with $n$-th sample, $\pi_n(B \mid D^n)$, is obtained via Bayes theorem as

$$\pi_n(B \mid \xi^n) = \frac{\int_B p_n(\xi^n \mid \theta)\, d\pi_n(\theta)}{\int_\Theta p_n(\xi^n \mid \theta)\, d\pi_n(\theta)}, \quad \text{where } B \subset \Theta,$$

as we assume that with increasing $n$, the data $\xi^n$ provide more information about $\theta \in \Theta$.

## A.1    Proof of Proposition 3.1

We start with the notion of posterior consistency. For simplicity let us redefine $\theta_{true} \equiv \theta_0$. Posterior consistency loosely means that for $n \to \infty$, the posterior $\pi\big(\theta|(\xi_t)^n\big)$ converges weakly to the Dirac measure $\delta_{\theta_0}$ in $\Theta$, and this depends on the topology of $\Theta$. More formally, let us assume that $\Theta$ is a metric space equipped with metric $d(\cdot, \cdot)$, then:

**Definition A.1** (Posterior Consistency)**.** Given a prior distribution $\pi(\theta) \in \Pi$, a posterior distribution $\pi\big(\theta|(\xi_t)^n\big)$ is said to be consistent w.r.t. a true parameter $\theta_0 \in (\Theta, d)$ if

$$\pi\big(\theta \in \Theta : d(\theta, \theta_0) > \epsilon \mid \xi_t^n\big) \overset{P_{\theta_0}}{\to} 0 \quad \text{as} \quad n \to \infty.$$

One important consequence of posterior consistency that we are going to make use of relates to the consistency of estimators $\hat{\theta}_n = T(\xi^n)$ defined from the posterior, and it is stated as follows:

**Proposition A.2** (Estimators Consistency)**.** *Assume $\Theta^* \subset \Theta$ is a subset such that posterior consistency holds for $\theta_0 \in \Theta^*$. The following facts are true:*

  *i) There exists $\exists \hat{\theta}_n = T(\xi^n)$ an estimator that is consistent for $\theta_0 \in \Theta^*$, i.e., $d(\hat{\theta}_n, \theta_0) \overset{P_{\theta_0}}{\to} 0$ when $\xi^n \sim p(\xi^n|\theta_0)$*

  *ii) If $\Theta$ is convex and $d(\cdot, \cdot)$ is a bounded and convex distance function, then $\hat{\theta}_n = T(\xi^n)$ can be the posterior mean $\hat{\theta}_n = T(\xi^n) = \int \theta\, d\pi_n(\theta|\xi^n)$*

  *iii) If $g : \Theta \to \Theta$ is a function continuous at $\theta_0$, then $d\big(g(\hat{\theta}_n), g(\theta_0)\big) \overset{P_{\theta_0}}{\to} 0$.*

Convexity of $\Theta$ indeed holds for the non-parametric regression problem defined by $Y_i = \theta(X_i) + \varepsilon_i$, where $f = \theta \in \Theta = C(\mathcal{X})$ and $X \in \mathcal{X}$. As for what requirements are needed for posterior consistency, we need intuitively that prior $\pi(\theta)$ do not exclude $\theta_0$ from its support. Define the 'model' as the likelihood density function that generates samples $\xi_t^n \sim p_0 \in \mathcal{P}$, where $\mathcal{P}$ is a probability measure, and $p_0 \in \mathcal{P}$ being the true density. Schwartz (1965) derived conditions for posterior consistency based on whether the true data generating model $p_0$ belongs to the KL support of the prior $\pi$. In particular, we define

**Definition A.3** (KL-support of $p(\theta)$)**.** By denoting as $d_{KL}(p, q)$ the KL-divergence between distributions $p$ and $q$, $p_0$ is said to belong to the KL support of the prior $\pi$, written $p_0 \in \mathrm{KL}(\pi)$, if

$$\forall \epsilon > 0, \quad \pi\big(p : d_{KL}(p_0, p) < \epsilon\big) > 0.$$

Now, define $U \subset \mathcal{P}$ as a open neighborhood of $p_0$ according to metric $d(\cdot, \cdot)$, then we intuitively have that the posterior $\pi(\cdot|\xi_t^n)$ is also consistent if and only if for every open neighborhood $U$ of $p_0$, $\pi(U^c|\xi_t^n) \to 0$ (i.e., if all neighborhood around $p_0$ collapse to 0). This is formalized as follows (Schwartz, 1965; Ghosal & Van der Vaart, 2017). Given a neighborhood $U \subset \mathcal{P}$, then we can test hypotheses $H_0 : p = p_0$ versus $H_1 : p \in U$. Assume these exist a real function $\varphi_n = \varphi(\xi_1, ..., \xi_n) : \Xi \to [0, 1]$, representing the probability of rejecting $H_0$, such that $\mathbb{E}_{p_0}[\varphi_n] \to 0$ and $\sup_{p \in U} \mathbb{E}_p[1 - \varphi_n] \to 0$ (i.e., probability of rejecting $H_0$ goes to zero if $p = p_0$, and conversely probability of rejecting $H_1$ goes to zero when $p \neq p_0$). The idea of Schwartz (1965) theorem is that posterior consistency is guaranteed if the prior $\pi$ assigns mass that is 'arbitrarily close' to the true model $p_0 \in \mathcal{P}$ and as $n$ grows we can more correctly classify $H_0$ vs $H_1$. In addition, define $P_0^\infty$ as the joint density of $\xi^n = (\xi_1, \xi_2, ...)$ under data generating model $p_0$. In full form then the weak consistency theorem reads:

**Theorem A.4** (Schwartz (1965)). *Assume that $p_0 \in KL(\pi)$, and that for neighborhoods $U_n \subset \mathcal{P}$ of $p_0$ there are test functions $\varphi_n$ satisfying the following requirements:*

$$\mathbb{E}_{p_0} \varphi_n \leq B e^{-bn}, \quad \sup_{p \in U_n^c} \mathbb{E}_p (1 - \varphi_n) \leq B e^{-bn}$$

*for some constants $b, B > 0$, then $\pi(U_n|\xi^n) \overset{P_0^\infty}{\to} 0$, or equivalently, $\pi(\theta|\xi_t^n) \overset{P_0^\infty}{\to} \delta_{\theta_0}$ via corollary result.*

Now we revert back to our specific dynamics model case, having true parameters $\theta_0 = (f_0, \sigma_0^2) \in (\Theta, d)$. Assuming the data generating model $p_0$ described by equation (3), and assuming that $p_0 \in KL(\pi)$, we have by Schwartz's theorem that $\pi(\theta|\xi_t^n) \overset{P_{\theta_0}}{\to} \delta_{\theta_0}$. As stated in Proposition 2.1 this translate into

$$p\big(\theta \in \Theta : d\big((f, \sigma), (f_0, \sigma_0)\big) > \epsilon \mid (s, a)^n\big) \overset{P_{\theta_0}}{\to} 0 \quad \text{as} \quad n \to \infty,$$

in our case. As per Proposition A.2 reported above then we know that we can construct an estimator $g(\hat{\theta}_n) = T(\xi^n)$ from the posterior $\pi(\theta|\xi^n))$ that is consistent for the parameter $g(\theta_0) \in \Theta$. Arbitrarily, we can pick $g(\hat{\theta}_n)$ to be exactly $T(\xi^n) = \text{IG}_\theta(\xi^n, s_{t+1})$. Then, according to Proposition A.2, we have $d\big(\text{IG}_\theta(\xi^n, s_{t+1}), \text{IG}_{\theta_0}(\xi^n, s_{t+1})\big) \overset{P_{\theta_0}}{\to} 0$ or

$$Pr\big(\theta \in \Theta : d\big(\text{IG}_\theta(\xi^n, s_{t+1}), \text{IG}_{\theta_0}(\xi^n, s_{t+1})\big) > \epsilon\big) \overset{P_{\theta_0}}{\to} 0 \quad \text{as} \quad n \to \infty.$$

However, since at convergence we have $\pi(\theta|\xi_t^n) \overset{P_0^\infty}{\to} \delta_{\theta_0}$, then the equivalence $p(\theta|\xi_t) = p(\theta|\xi_t, s_{t+1})$ holds (i.e., new datum $s_{t+1}$ does not convey any additional information on $\theta \in \Theta$). This implies that $\text{IG}_{\theta_0}(\xi^n, s_{t+1})) = H[\delta_{\theta_0}] - H[\delta_{\theta_0}] = 0$, thus the convergence result is $H[p(\theta|\xi_t)] - H[p(\theta|\xi_t, s_{t+1})] \overset{P_{\theta_0}}{\to} H[\delta_{\theta_0}] - H[\delta_{\theta_0}] = 0$, or similarly $d\big(\text{IG}_\theta(\xi^n, s_{t+1}), 0\big) \overset{P_{\theta_0}}{\to} 0$.

### A.1.1 Convergence to Optimal Value Function

As a corollary of Proposition 3.1 proved above, we can straightforwardly show that the value function $V^{*,augm}$ defined by the augmented rewards $r_t = r_t^e + \eta_i r_t^i$ is such that $V^{*,augm} \overset{P}{\to} V^*$. More formally, we define $V^{*,augm}$ as follows

$$V_t^{*,augm}(s, \theta) = \max_{a \in \mathcal{A}} \left[ r^e(s, a) + \eta_i r^i(s, a; \theta) + \gamma \int_{\mathcal{S}, \Theta} p(s'|s, \theta, a) V_{t-1}^*(s', \theta') ds' d\theta' \right] \tag{7}$$

while the optimal value function associated with the original MDP's extrinsic rewards only is simplified to

$$V_t^*(s; \theta_0) = \max_{a \in \mathcal{A}} \left[ r^e(s, a) + \gamma \int_{\mathcal{S}} p(s'|s, a; \theta_0) V_{t-1}^*(s'; \theta_0) ds' \right]$$

Wrapping this up in a corollary statement:

**Corollary A.5** ($V^{*,augm}$ Convergence). *Under the conditions for posterior consistency of Proposition 2.1, it is easy to see from Eq. (7) that we have $V_t^{*,augm}(s, \theta) \overset{P_{\theta_0}}{\to} V_t^*(s; \theta_0)$, as $r_t^i = \text{IG}_\theta(\cdot) \overset{P_{\theta_0}}{\to} 0$ and $\theta \overset{P_{\theta_0}}{\to} \theta_0$.*

## A.2 Proof of Proposition 3.3

The intuition behind Proposition 3.3 is simply that, following the statement of Proposition 3.1, if we can prove a contraction rate $\epsilon_n$ holds for posterior convergence $\pi(\theta|\xi^n) \to \delta_{\theta_0}$, then in light of the proves in the above section, both $d\big(\mathrm{IG}_\theta(\xi^n, s_{t+1}), 0\big) \overset{P_{\theta_0}}{\to} 0$ and $d(V^{augm,*}, V^*) \overset{P_{\theta_0}}{\to} 0$ happen at the same rate $\epsilon_n$. To prove Proposition 3.3, we begin by defining contraction rates for a consistent posterior $\pi(\theta|\xi_t^n)$, where the true $\theta_0 \in (\Theta, d)$. Notice that posterior consistency always implies a contraction rate $\epsilon_n$ (Ghosal & Van der Vaart, 2017). We define

**Definition A.6** (Contraction Rate). A posterior $\pi(\theta|\xi_t^n)$ is said to contract to $\delta_{\theta_0}$ at the rate $\epsilon_n \to 0$, if for every constant $\forall M > 0$:

$$\pi\big(\theta \in \Theta : d(\theta, \theta_0) > M\epsilon_n \mid \xi_n\big) \overset{P_{\theta_0}}{\to} 0 \quad \text{when} \quad \xi_n \sim p_n(\xi|\theta_0) \,.$$

A corollary of Proposition A.2 then straightforwardly holds, with respect to posterior estimators $\hat{\theta}_n = T(\xi^n)$ convergence rate, and states:

**Corollary A.7** (Estimator Contraction). *If posterior $\pi(\theta|\xi_t^n)$ contracts at rate $\epsilon_n$ (or faster) to $\theta_0 \in \Theta_0 \subset \Theta$, then there exists an estimator $\hat{\theta}_n = T(\xi^n)$ that is consistent for $\theta \in \Theta_0$ and converges at least as fast as $\epsilon_n$.*

Assume now the dynamics model presented in Section 3.2 holds, that is:

$$S_{t+1} = f(S_t, A_t) + \varepsilon_t \,, \quad \text{where} \quad \varepsilon_t \sim \mathcal{N}(0, \sigma^2) \,, \text{ and } \sigma^2 \in \mathbb{R}^+$$

and assume that $(f_0, \sigma_0) \in \mathcal{C}^\alpha(\mathcal{X}) \times [c, d]$, where: $\mathcal{X} = \mathcal{S} \times \mathcal{A}$ is a compact subset of $\mathbb{R}$; $\mathcal{C}^\alpha(\cdot)$ is the class of continuous functions with finite Hölder norm of order $\alpha$; and $[c, d] \subset \mathbb{R}^+$. Notice that we assumed that $\mathcal{S}, \mathcal{A} \subseteq \mathbb{R}$, so that the target variable $s_{t+1}$ is single-task continuous $S_{t+1} \in \mathbb{R}$, but the following would hold also for multi-task settings, just individually and independently for each task. Finally, assume that $f \sim \mathcal{GP}\big(0, C(\cdot, \cdot)\big)$, a GP prior where $C(\cdot, \cdot)$ is the isotropic squared exponential kernel, i.e., $C(x, y) = \exp\{-l^2\|x - y\|^2\}$ with length parameter $l \in \mathbb{R}^+$. Then, one can prove that posterior $\pi(f, \sigma|\xi^n)$ contracts at the optimal minimax rate up to a log constant, that is:

**Theorem A.8** (van der Vaart & van Zanten (2008)). *Assume $(f_0, \sigma_0) \in \mathcal{C}^\alpha(\mathcal{X}) \times [c, d]$, where $\mathcal{X}$ is a compact subset of $\mathbb{R}$, the dynamics model in (3), and $f \sim \mathcal{GP}\big(0, C(\cdot, \cdot)\big)$ prior where $C(\cdot, \cdot)$ is the squared exponential kernel. Then, denoting $p = |\mathcal{S} \times \mathcal{A}|$:*

$$\pi\left(\{(f, \sigma) : d\big((f.\sigma), (f_0, \sigma_0)\big) > \epsilon_n\} \mid \xi_t^n\right) \overset{P_0^\infty}{\to} 0 \quad as \quad n \to \infty$$

*where $\epsilon_n = n^{-\frac{1}{(2+p/\alpha)}}(\log n)^t$ with $t = 1 - \frac{1}{(2+4\alpha/p)}$.*

Notice that with $(\log n)^t = 1$, the above is equal to the minimax rate (best rate of estimation) for functions in the class $\mathcal{C}^\alpha(\mathcal{X})$ (Yang & Barron, 1999). Since posterior consistency implies the existence of a contraction rate $\epsilon_n$, then given that conditions for posterior consistency hold in Proposition 3.1, and given result in Corollary A.7, we have that $\mathrm{IG}_\theta(\xi^n, s_{t+1}) \to 0$ at the same rate $\epsilon_n$. Same hold for $V_t^{*,augm}(s, \theta) \to V_t^*(s; \theta_0)$, by extending Corollary A.5. Thus, this implies that under the condition of Theorem A.8 above, we have that $\mathrm{IG}_\theta(\xi^n, s_{t+1}) \to 0$ and $V_t^{*,augm}(s, \theta) \to V_t^*(s; \theta_0)$ at same rate $\epsilon_n = n^{-\frac{1}{(2+p/\alpha)}}(\log n)^t$.

Moreover, Van Der Vaart & Van Zanten (2011) show that the optimal minimax rate $\epsilon_n = n^{-\frac{1}{(2+p/\alpha)}}$ (Yang & Barron, 1999) for posterior contraction can be also achieve instead by imposing a further restriction on $f_0$ and assuming the covariance function $C(\cdot, \cdot|\cdot)$ is a Matérn kernel. Define $H^\alpha(\mathcal{X})$ as the Sobolev space, then:

**Theorem A.9** (Van Der Vaart & Van Zanten (2011)). *Given $f_0 \in C^\alpha(\mathcal{X}) \cap H^\alpha(\mathcal{X})$ and $f \sim \mathcal{GP}\big(0, C(\cdot, \cdot|\cdot)\big)$ where $C(\cdot, \cdot|\cdot)$ is a Matérn kernel on $\mathcal{X} = [0, 1]^p$, then the posterior $\pi(f|\cdot)$ contracts at rate $\epsilon_n = n^{-\frac{1}{(2+p/\alpha)}}$.*

Thus, under the assumptions specified by Theorem A.9, also $\mathrm{IG}_\theta(\xi^n, s_{t+1}) \to 0$ and $V_t^{*,augm}(s, \theta) \to V_t^*(s; \theta_0)$ happen at the same rate $\epsilon_n = n^{-\frac{1}{(2+p/\alpha)}}$.

### A.3 Brief Discussion on Active vs Passive Learning Rates

The posterior contraction rates $\epsilon_n$ derived in the results above all pertains to the classical case of 'passive' learning (Yang & Barron, 1999). We note that these can be improved under some conditions with an active sampling strategy (Hanneke et al., 2014; Hanneke & Yang, 2015), which is what the work ultimately advocates for the realm of data-efficient exploration in MBRL. In Willett et al. (2005), the authors show that the minimax convergence rates for regression cases where $f_0$ is a piecewise constant function can be strictly reduced from $\epsilon_n = n^{-1/|\mathcal{X}|}$ (passive) to $\epsilon_n = n^{-1/(|\mathcal{X}|-1)}$ (active). Castro & Nowak (2008) instead show that for classification problems with similar Hölder smooth decision boundaries the minimax lower bound convergence rate can be tightly improved from $\epsilon_n = n^{\kappa/(\kappa+\rho-1)}$ (passive) to $\epsilon_n = n^{\kappa/(\kappa+\rho-2)}$ (active), where $\rho = (|\mathcal{X}|-1)/\alpha$ and $\kappa$ is the highest integer such that $\kappa < \alpha$. As a final example, results in Wang et al. (2018) show that if $f_0$ is strongly smooth and convex (e.g., as in Example 2 in their paper), with $\alpha = 2$, one can achieve a much better rate of $\epsilon_n = n^{-1/2}$ compared to the passive minimax, which in that example's case is $\epsilon_n = n^{-2/4+|\mathcal{X}|}$.

## B Information Gain Derivations

In this second appendix section we include the full derivations of the Expected Information Gain (EIG) measures (Lindley, 1956; Bernardo, 1979; Rainforth et al., 2024) and the version of EIG under a Gaussian dynamics model and under a deep ensemble dynamics model. We assume the setup is the one described in the MDP definition in the main paper. The Information Gain (IG) of the full transition $t \to t+1$ composed by $(s_t, a_t, s_{t+1})$ is defined as

$$\mathrm{IG}_\theta(s_t, a_t, s_{t+1}) = H[p(\theta|s_t, a_t)] - H[p(\theta|s_t, a_t, s_{t+1})] = \tag{8}$$
$$= \mathbb{E}_{p(\theta|s_t, a_t, s_{t+1})}[\log p(\theta|s_t, a_t, s_{t+1})] - \mathbb{E}_{p(\theta|s_t, a_t)}[\log p(\theta|s_t, a_t)],$$

where $p(\theta|s_{t+1}, s_t, a_t) \propto p(\theta)p(s_{t+1}|\theta, s_t, a_t)$, and where $H[p(x)] = \mathbb{E}[-\log p(x)]$ is the Shannon entropy (Shannon, 1948) and $\theta \in \Theta$ the set of dynamics parameters. As stated in main paper, the issue associated with computing $\mathrm{IG}_\theta(s_t, a_t, s_{t+1})$ is that it can be done only in a reactive setting where $s_{t+1}$ is actually revealed to the agent. Thus in an active setting, we have to resort to an expected value surrogate version of it, $\mathrm{EIG}_\theta(\cdot)$.

### B.1 Expected Information Gain

As $s_{t+1}$ is not revealed to the agent at time $t$, we can use $\mathrm{EIG}_\theta(s_t, a_t) = \mathbb{E}_{p_\theta(s_{t+1}|s_t, a_t)}\big[\mathrm{IG}_\theta(s_t, a_t, s_{t+1})\big]$, which essentially marginalizes over possible next $s_{t+1} \in \mathcal{S}$, defined in full form as:

$$\mathrm{EIG}_\theta(s_t, a_t) = \mathbb{E}_{p(s_{t+1}|s_t, a_t)}\big[\mathrm{IG}_\theta(s_t, a_t, s_{t+1})\big] =$$
$$= \mathbb{E}_{p(s_{t+1}|s_t, a_t)}\big[\mathbb{E}_{p(\theta|s_t, a_t, s_{t+1})}[\log p(\theta|s_t, a_t, s_{t+1})] - \mathbb{E}_{p(\theta|s_t, a_t)}[\log p(\theta|s_t, a_t)]\big]. \tag{9}$$

Now we note that we can re-write $p(s_{t+1}|s_t, a_t) = \int p(s_{t+1}|s_t, a_t, \theta)p(\theta|s_t, a_t)\, d\theta$ by marginalization, and that following Bayes theorem:

$$p(\theta|s_t, a_t) = \frac{p(s_{t+1}|s_t, a_t, \theta)p(s_t, a_t, \theta)}{\int_\Theta p(s_{t+1}|s_t, a_t, \theta)p(s_t, a_t, \theta)\, d\theta} =$$
$$= \frac{p(s_{t+1}|s_t, a_t, \theta)p(s_t, a_t|\theta)p(\theta)}{\int_\Theta p(s_{t+1}|s_t, a_t, \theta)p(s_t, a_t|\theta)p(\theta)\, d\theta}.$$

Using these two facts and applying them to (9) we get:

$$\mathbb{E}_{p(s_{t+1}|s_t,a_t)}\left[\mathbb{E}_{p(\theta|s_t,a_t,s_{t+1})}[\log p(\theta|s_t,a_t,s_{t+1})] - \mathbb{E}_{p(\theta|s_t,a_t)}[\log p(\theta|s_t,a_t)]\right] =$$

$$= \mathbb{E}_{p(\theta)p(s_{t+1}|\theta,s_t,a_t)}\left[\log \frac{\frac{p(s_{t+1}|s_t,a_t,\theta)p(s_t,a_t|\theta)p(\theta)}{\int_\Theta p(s_{t+1}|s_t,a_t,\theta)p(s_t,a_t|\theta)p(\theta)\,d\theta}}{\frac{p(s_t,a_t|\theta)p(\theta)}{p(s_t,a_t)}}\right] =$$

$$= \mathbb{E}_{p(\theta)p(s_{t+1}|\theta,s_t,a_t)}\left[\log \frac{\frac{p(s_{t+1}|s_t,a_t,\theta)p(s_t,a_t|\theta)p(\theta)}{p(s_{t+1}|s_t,a_t)p(s_t,a_t)}}{\frac{p(s_t,a_t|\theta)p(\theta)}{p(s_t,a_t)}}\right] =$$

$$= \mathbb{E}_{p(\theta)p(s_{t+1}|s_t,a_t,\theta)}\left[\log p(s_{t+1}|s_t,a_t,\theta) - \log p(s_{t+1}|s_t,a_t)\right] =$$

$$= H\left[p(s_{t+1}|s_t,a_t)\right] - \mathbb{E}_{p(\theta|s_t,a_t)}\left[H[p(s_{t+1}|s_t,a_t;\theta)]\right] =$$

$$= \mathbb{E}_{p(\theta|s_t,a_t)}[D_{KL}(p(s_{t+1}|s_t,a_t;\theta)\|p(s_{t+1}|s_t,a_t))]\ ,$$

by canceling terms out and re-ordering. Thus, we have now obtained an equivalent specification of $\text{EIG}_\theta(s_t,a_t)$ that can be computed using only the posterior predictive distribution $p_\theta(s_{t+1}|s_t,a_t)$, which marginalizes over parameters $\theta \in \Theta$ according to their posterior, i.e., $p_\theta(s_{t+1}|s_t,a_t) = \int_\Theta p_\theta(s_{t+1}|s_t,a_t,\theta)p(\theta|s_t,a_t)\,d\theta$. $\text{EIG}_\theta(s_t,a_t)$ in this form can be interpreted as follows: it represents the expected reduction in the predictive uncertainty over the next state $s_{t+1}$ obtained from observing a different set of parameters $\theta$.

## B.2 EIG under Gaussian dynamics

If the predictive posterior distribution is a (multivariate) Gaussian, such as in the case of Gaussian Processes and Deep Kernels, we can simplify $\text{EIG}_\theta(s_t,a_t)$ calculations even more. This is because if $p(\mathbf{x}) \triangleq \mathcal{N}(\mu,\Sigma)$, the entropy $H[p(\mathbf{x})]$ can be simplified (Cover, 1999) as:

$$H[p(\mathbf{x})] = -\int_{-\infty}^\infty \mathcal{N}(\mu,\Sigma)\,\log\mathcal{N}(\mu,\Sigma)\,d\mathbf{x} =$$

$$= \frac{D}{2}\log 2\pi + \frac{1}{2}\log\det(\Sigma) + \frac{1}{2}\mathbb{E}\left[(\mathbf{x}-\mu)^\top\Sigma^{-1}(\mathbf{x}-\mu)\right] =$$

$$= \frac{D}{2}\log 2\pi + \frac{1}{2}\log\det(\Sigma) + \frac{1}{2}D =$$

$$= \frac{1}{2}\log\det(\Sigma) + \frac{D}{2}(1 + \log 2\pi)\ ,$$

where $D$ is the dimensionality of the multivariate normal, that in our case corresponds to $D = |\mathcal{S}|$, and where the term $\mathbb{E}\left[(\mathbf{x}-\mu)^\top\Sigma^{-1}(\mathbf{x}-\mu)\right]$ is simplified as follows:

$$\mathbb{E}\left[(\mathbf{x}-\mu)^\top\Sigma^{-1}(\mathbf{x}-\mu)\right] = \mathbb{E}\left[tr((\mathbf{x}-\mu)^\top\Sigma^{-1}(\mathbf{x}-\mu))\right] =$$

$$= \mathbb{E}\left[tr(\Sigma^{-1}(\mathbf{x}-\mu)(\mathbf{x}-\mu)^\top)\right] =$$

$$= tr(\mathbb{E}\left[\Sigma^{-1}(\mathbf{x}-\mu)^\top(\mathbf{x}-\mu)\right]) =$$

$$= tr(\Sigma^{-1}\Sigma) =$$

$$= tr(I_D) =$$

$$= D$$

Taking this simplification into account, we can in turn reduce $\text{EIG}_\theta(s_t,a_t)$ calculations to the following:

$$\mathbb{E}_{p(\theta)p(s_{t+1}|s_t,a_t,\theta)}\left[H[p(s_{t+1}|s_t,a_t)] - H[p(s_{t+1}|s_t,a_t,\theta)]\right] =$$

$$= \mathbb{E}_{p(\theta)p(s_{t+1}|s_t,a_t,\theta)}\left[\frac{1}{2}\log\det(\mathbb{V}[p(s_{t+1}|s_t,a_t)]) - \frac{1}{2}\log\det(\mathbb{V}[p(s_{t+1}|s_t,a_t,\theta)])\right] =$$

$$= \frac{1}{2}\Big(\log\det(\mathbb{V}[p(s_{t+1}|s_t,a_t)]) - \mathbb{E}_{p(\theta)}\left[\log\det(\mathbb{V}[p(s_{t+1}|s_t,a_t,\theta)])\right]\Big)\ ,$$

where $\mathbb{V}[p(\cdot)]$ denotes the variance of distribution $p(\cdot)$.

### B.3 EIG as Disagreement of a Gaussian Mixture

In settings where the dynamics model being employed does not yield a full, Gaussian posterior predictive distribution such as in the case of GPs and DKs, then closed-forms EIG as described above cannot really be computed. This is, for example, the case where deep ensembles are utilized as a model for the dynamics. However, we can show that EIG in this setting can be computed as the Jensen Shannon-Divergence among a Gaussian mixture consisting of all the $M$ probabilistic neural network predictive distributions in an ensemble, i.e., $\{p(r, s'|s, a; \theta_m)\}_{m=1}^M$. This is derived as follows. First consider that:

$$\text{EIG}_\theta(s_t, a_t) = \mathbb{E}_{p(\theta|s_t, a_t)p(s_{t+1}|s_t, a_t; \theta)}\big[\log p(s_{t+1}|s_t, a_t; \theta) - \log p(s_{t+1}|s_t, a_t)\big] =$$

$$= \mathbb{E}_{p(\theta|s_t, a_t)p(s_{t+1}|s_t, a_t; \theta)}\Big[\log p(s_{t+1}|s_t, a_t; \theta) - \log \int_\Theta p(s_{t+1}|s_t, a_t; \theta)\, d\theta\Big] =$$

$$= H\big[\mathbb{E}_{p(\theta|s_t, a_t)}[p(s_{t+1}|s_t, a_t; \theta)]\big] - \mathbb{E}_{p(\theta|s_t, a_t)}\big[H[p(s_{t+1}|s_t, a_t; \theta)]\big] \,,$$

where the term $\mathbb{E}_{p(\theta|s_t, a_t)}[p(s_{t+1}|s_t, a_t; \theta)]$ is the expected predictive distribution over next states, which equals the marginal $p(s_{t+1}|s_t, a_t)$ in the previous section's specification. Thus, $\mathbb{E}_{p(\theta|s_t, a_t)}[p(s_{t+1}|s_t, a_t; \theta)] = H[p(s_{t+1}|s_t, a_t)]$. The other term $\mathbb{E}_{p(\theta|s_t, a_t)}\big[H[p(s_{t+1}|s_t, a_t; \theta)]\big]$ is the expected entropy of the predictive distribution for specific $\theta$ values. Now, note that the JSD for a set of $M$ distributions with discrete weights $\pi_m$ is defined as:

$$\text{JSD}_{\pi_1, ..., \pi_M}(p_1, ..., p_M) = H\left[\sum_{m=1}^M \pi_m p_m\right] - \sum_{m=1}^M \pi_m H[p_m] \,.$$

In our context, we can view the probability $p(\theta|s_t, a_t)$ as continuous weights, so that we obtain:

$$\text{EIG}_\theta(s_t, a_t) = H\big[\mathbb{E}_{p(\theta|s_t, a_t)}[p(s_{t+1}|s_t, a_t; \theta)]\big] - \mathbb{E}_{p(\theta|s_t, a_t)}\big[H[p(s_{t+1}|s_t, a_t; \theta)]\big] =$$

$$= \text{JSD}_\theta\big(p(s_{t+1}|s_t, a_t; \theta) \mid \theta \sim p(\theta|s_t, a_t)\big) \,.$$

## C  Additional Information on the PTS-BE algorithm

This section includes some additional information regarding the components featuring in the PTS-BE algorithm that we introduce in the main paper. In particular, we describe the specifications of all the different dynamics model that we tried in synergy with the proposed PTS-BE algorithm, namely Exact and Sparse Input versions of both Gaussian Processes and Deep Kernels, and then Deep Ensembles.

### C.1  Models for the Environment Dynamics

The principal component in both a "BE" type of algorithm, that employ retrospective intrinsic reward augmentation, and a PTS-BE one, that is based on the planning-to-explore principle instead, is a Bayesian model of the environment dynamics. This is because only Bayesian, uncertainty aware models allow for the incorporation of information gain measures. We review specifications of three classes of regression models: (Stochastic Variational) Gaussian Processes (Quinonero-Candela & Rasmussen, 2005; Rasmussen et al., 2006; Hensman et al., 2013), Deep Kernels (Wilson et al., 2016b;a) and deep ensembles (Lakshminarayanan et al., 2017; Rahaman et al., 2021). We briefly discuss their specific implications in terms of posterior coverage uncertainty properties.

#### C.1.1  Gaussian Processes

Gaussian Process (GP) Regression models place a prior on the functional form of $f(\cdot) \sim \mathcal{GP}\big(m(\cdot), k(\cdot, \cdot)\big)$, such that $p(f) \triangleq \mathcal{N}(f \mid m(\cdot), k(\cdot, \cdot))$, where $m(\cdot)$ is a mean function (e.g., constant, linear, etc.) and $k(\cdot, \cdot)$ a kernel function (e.g., linear, squared-exponential, matérn, etc.). Prior knowledge about $f \in \mathcal{F}$ can be efficiently conveyed through the choice of $m(\cdot)$ and especially $k(\cdot, \cdot)$ (e.g., domain knowledge about the physics of the environment), which governs main features of the function approximator $\hat{f}(\cdot)$ such as smoothness and sparsity. This feature makes it appealing in many applications where knowledge about the environment can

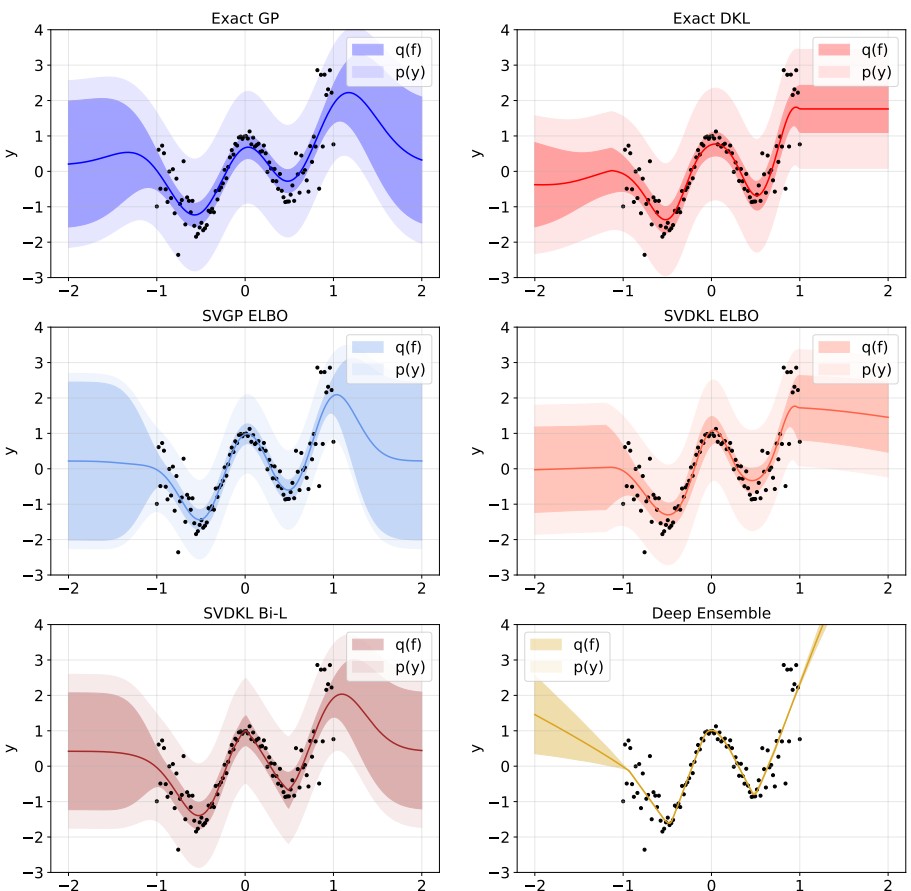

Figure 7: Comparison of uncertainty quantification models on a simple one dimensional regression example. The sample size of generated data points is $n = 100$. The models considered are, starting from the top left corner towards the bottom right corner: i) Exact GP, trained on the whole sample; ii) Exact DKL, trained on the whole sample; iii) Stochastic Variational GP, trained on the ELBO loss over a subset of 20 data points; iv) SV DKL, trained the same way as SVGP; v) SV DKL coupled with dropout fully-connected ResNet structure and Bi-Lipschitz constraints to avoid feature collapse; vi) Deep ensembles made of 5 neural networks models.

be incorporated a priori. The joint density of $(\mathbf{y}, \mathbf{f})$ and the marginal likelihood, which is used for training, in a GP take the following form

$$p(\mathbf{y}, \mathbf{f}|X) = p(\mathbf{y}|\mathbf{f}, \sigma^2)p\big(\mathbf{f}|X; m(\cdot), k(\cdot, \cdot)\big) \quad \text{and} \quad p(\mathbf{y}|X) = \int_{\mathcal{F}} p(\mathbf{y}|\mathbf{f}, \sigma^2)p(\mathbf{f}|X)\,d\mathbf{f}\ ,$$

where all the quantities of interest can be computed analytically as the distributions are Gaussian. From this point of view, GPs have the advantage of being complementary to the simplified version of $\mathrm{EIG}_\theta(s_t, a_t)$ derived above, as their predictive posterior distribution on $s_{t+1}$ is in fact Gaussian as well. Besides GPs are well-known for their excellent uncertainty quantification properties. Their main drawback lies in their poor scalability, which is why we consider a Stochastic Variational approach where we learn a sub-sample of inducing points $\mathbf{u} = \{u_i\}_{i=1}^m$ (Hensman et al., 2013). We also considered using multitask/multioutput kernel learning (Alvarez et al., 2012) but did not notice any significant improvements, while training costs were higher for the higher number of parameters involved in the multi-task kernels.

Throughout the experiments presented in the work, we utilize a GP with constant prior mean function, $m(\cdot) = 0$ and with base squared exponential kernel $k(x, y) = C_{ff}(x, y) = \exp\{-l^2\|x - y\|^2\}$.

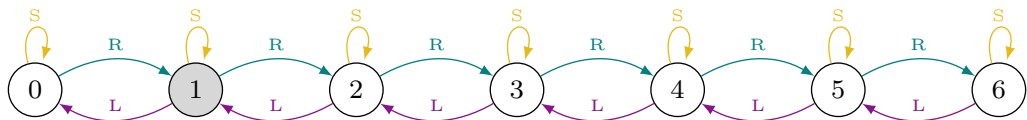

Figure 8: Example of unichain environment made of $L = 7$ states. Arrows represent the possible actions in each state, state one is shaded to indicate that the agent is spawned there as $s_0 = 1$.

### C.1.2 Deep Kernels

Deep kernels (Wilson et al., 2016b;a) are a generalization of GPs where inputs $(s_t, a_t)$, or just $s_t$, are first mapped to a (potentially lower dimensional) latent representation space $h_t$, $f_h : \mathcal{S} \times \mathcal{A} \to \mathcal{H}$ or $f_h : \mathcal{S} \to \mathcal{H}$, through a deep learning architecture. Then, the function to predict the next state, $f_s : \mathcal{H} \to \mathcal{S}$ or $f_s : \mathcal{H} \times \mathcal{A} \to \mathcal{S}$, can be assigned any GP prior (e.g., SVGP). The advantage of using deep kernels lies in the fact that they are better suited for dealing with high-dimensional state-action spaces thanks to their deep architecture, while they retain good function approximation and uncertainty quantification properties of GPs. On this matter, we specifically employ a version of deep kernels that uses a fully-connected ResNet neural net architecture (with dropout) for $f_h(\cdot)$ and incorporates a bi-Lipschitz constraint on the transformations in $f_h(\cdot)$ (van Amersfoort et al., 2021), in order to avoid the tendency of neural nets of exhibiting feature collapse (Guo et al., 2017). Note that the last GP layer in a deep kernel model scales by construction at a $\mathcal{O}(m^3|\mathcal{H}|^3)$ training and $\mathcal{O}(m^2|\mathcal{H}|^2)$ test cost, where $|\mathcal{H}|$ is directly controllable. Lastly, notice that a (SV) deep kernel still allow one to use the simplified version of $\text{EIG}_\theta(s_t, a_t)$ on the latent space $\mathcal{H}$, i.e., $\text{EIG}_\theta(h_t)$, as $p(s_{t+1}|h_t)$ is modelled as a Gaussian, as any component coming after the neural network block that models $p(s_{t+1}|h_t)$ behaves exactly as a Gaussian Process.

We have also considered implementing the MC dropout (Gal & Ghahramani, 2016) technique and apply it to the deep neural network layers of deep kernels in order to get better generalization properties, but did not notice any further improvement. MC Dropout consists in re-sampling a pre-trained neural network with dropout layers $K$ times, at test time, such that each prediction $\{(\mathbf{y}|X)\}_{i=1}^K$ represents a sample from the predictive distribution $q(\cdot \mid \cdot)$, with different weights $\omega = \{W\}_{i=1}^L$ and biases $\mathbf{b} = \{b\}_{i=1}^L$:

$$q(\mathbf{y}^* \mid \mathbf{x}^*) = \int p(\mathbf{y}^* \mid \mathbf{x}^*, \mathbf{W}, \mathbf{b}) \, p(\mathbf{W}, \mathbf{b}) \, d\mathbf{W}d\mathbf{b}.$$

MC Dropout approximates sampling from the Bayesian posterior predictive distribution, marginalizing out $\mathbf{W}, \mathbf{b}$. Perhaps the fact that MC dropout does not improve uncertainty quantification has to do with the fact that using a sub-sample of input points through the SV inducing points procedure already prevents the deep kernels model from being over-confident and collapsing to the mean (Rahaman et al., 2021), resulting into better generalization properties than standard deep kernels. However, we leave this topic to be investigated as part of future research.

Throughout the experiments, we consider a deep kernel model where the neural network architecture consists of two hidden layers of [32, 32] units, and a GP with constant mean and squared exponential kernel.

### C.1.3 Deep Ensembles

Deep ensembles (Lakshminarayanan et al., 2017) are arguably the most popular standard tool for uncertainty quantification in neural networks, thanks to their very good generalization properties compared to other sampling techniques such as MC dropout. The main difference with MC dropout is that deep ensembles requires to separately train (so they operate at train time) a pool of different neural network independently on the same data. Deep ensembles have been shown to have good out of sample uncertainty coverage (Wilson & Izmailov, 2020) in some examples, although they rarely match that of Gaussian Processes.

Some of the models, and their variations, considered for the environment dynamics are compared in the one dimensional regression example depicted in Figure 7. We use this simple example to depict each candidate model's generalization and uncertainty quantification properties. Description of each model is enclosed in the figure's caption. We can see how the best models emerging from this small scale study are the Exact

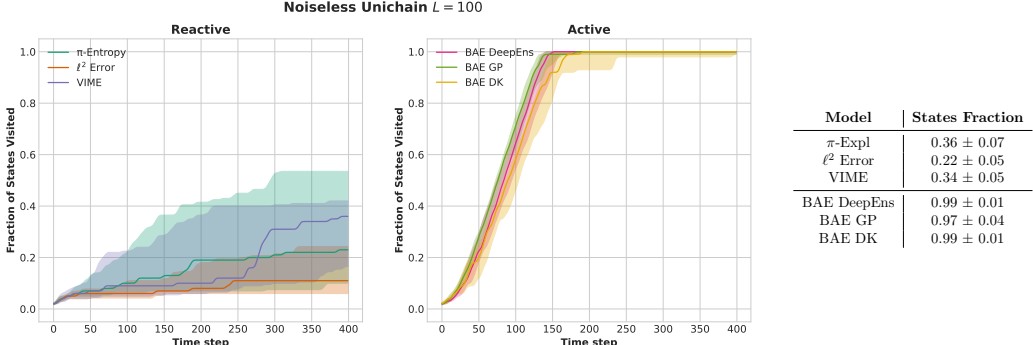

Figure 9: Results on the $L = 100$ states unichain environment exploration task. The plots show the median cumulative fraction of states visited (solid line) together with the 75th and 25th error bands, computed over 20 replications. The table instead reports the average fraction of states visited at termination (i.e., after 400 steps), together with the 95% confidence intervals.

GP, SVGP and SVDKL. In particular, SV DKL trains over 15 times faster than Exact DKL, and we notice that reducing the training sample to the 20 inducing points result in better uncertainty coverage in SV DKL versus Exact DKL.

Throughout the experiments in this work, we consider a deep ensemble of 5/7/10 neural networks, as we did not notice any appreciable improvement for ensembles larger than 10.

### C.1.4 Policy Specifications in PTS-BE

In the PTS-BE algorithm, a policy $\pi_\psi(s_t)$ parametrized by $\psi \in \Psi$ is update by using the model generated trajectories instead of the sampled ones to allow for planning and looking ahead. The policy $\pi_\psi(s_t)$ is typically parametrized by a neural network, and trained via gradient ascent, according to the policy gradient approach (Kakade, 2001). In the discrete action settings above, we typically make use of Proximal Policy Optimization (PPO) algorithms (Schulman et al., 2017). The standard PPO loss reads:

$$\mathcal{L}_t^{\text{PPO}}(\psi) = \mathbb{E}\big[\mathcal{L}_t^{\text{Clip}}(\psi) - c_1 \mathcal{L}_t^{\text{Value}}(\psi) + c_2 H[\pi_\psi(s_t)]\big]$$

where $\mathcal{L}_t^{\text{Clip}}$ is the clipped surrogate objective, $\mathcal{L}_t^{\text{Value}}$ is a squared loss on the value function $V(\cdot)$ and $H[\pi_\psi(s_t)]$ is a policy entropy term used to ensure sufficient exploration (for more details see Schulman et al. (2017)).

In the continuous action settings, we instead use Soft Actor-Critic (SAC) (Haarnoja et al., 2018). SAC updates the policy parameters by breaking down the policy improvement into separate Actor and Critic updates, according to the modified objective:

$$J(\pi) = \mathbb{E}_{\tau \sim \pi}\Big[\sum_t \gamma^t(r(s_t, a_t) + \alpha H(\pi(\cdot|s_t)))\Big] \,,$$

where the entropy term $(H(\pi(\cdot|s_t)))$ encourages exploration by keeping the policy as random as possible. The actor and critic networks, and the "temperature", are trained to minimize the following losses:

$$L(\psi_{actor}) = \mathbb{E}[\alpha * \log(\pi_\psi(a|s)) - Q_{\psi_{critic}}(s, a)] \,, \tag{10}$$

$$L(\psi_{critic}) = \mathbb{E}\Big[\Big(Q_{\psi_{critic}}(s, a) - \big(r + \gamma(Q_{target}(s', a') - \alpha \log \pi(a'|s'))\big)\Big)^2\Big] \,, \tag{11}$$

$$L(\alpha) = \mathbb{E}[-\alpha \log(\pi_{\psi_{actor}}(a|s)) - \alpha H_{target}] \,, \tag{12}$$

where the temperature parameter $\alpha$ is updated to achieve a target entropy $H_{target}$.

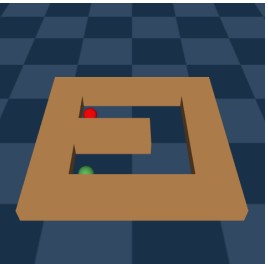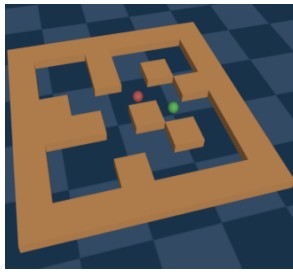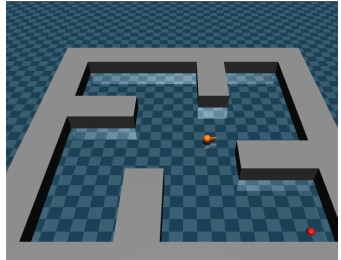

Figure 10: Visual 3D renditions of three types of mazes that can be obtained from the 2D Maze environments via customization of the maze structure: i) a u-shaped maze such as the one employed in the experiments above; ii) a medium sized maze with ledges and obstacles in the middle, again similar to the one emplyed in the experiment; iii) a larger maze with just ledges on the sides. The red ball denotes the goal object.

# D  Additional Experiments

## D.1  The Unichain Environment

In this last supplementary material section we include a brief description of the Unichain enviroment, plus additional results on it where we consider a longer chain of $L = 100$ states, without noise this time.

As outlined in the main paper, the unichain environment consists in a simple sequence of $L$ Markov sequentially connected states, where the goal is usually to explore all the possible states and reach the final one (Puterman, 2014; Osband et al., 2016). We consider a discrete action space defined as $\mathcal{A} = \{\text{go left}, \text{stay}, \text{go right}\}$. If the agent is in the first or last state and tries to go left or right, it automatically hit a wall and remains in that state. The agent is initially spawned in the second state $s_0 = 1$ (state counter starting from 0). The reward function $r^e : \mathcal{S} \times \mathcal{A} \to \mathbb{R}$ is sparse and assigns 0.001 to visiting the first state, 1 to visiting the last state, and 0 otherwise, i.e.:

$$r^e(s_t, a_t) = \begin{cases} 0.001 & \text{if } s_t = 0 \\ 1 & \text{if } s_t = L \\ 0 & \text{otherwise .} \end{cases}$$

The sub-optimal reward associated with state 0 represents a 'reward trap' for algorithms based solely on extrinsic reward exploitation, as it acts as disincentive for the agent to move elsewhere and explore. Thus, strong exploration bonus is needed to move it away from there. A simple visual representation of a $L = 7$ state unichain environment, together with the possible actions is depicted in Figure 8. The shaded state $s_0 = 1$ represents the spawning location.

## D.2  Additional Experiments on the 100 States Unichain Environment

In this subsection, we present the additional results on the $L = 100$ unichain environment, but without added noise this time. The task is again purely exploratory, and the methods compared are a subset of those compared in the main paper: i) PPO policy entropy $H(\pi_\psi(s_t))$ regularizer (Schulman et al., 2017) (**pi-Entropy**); ii) PPO with $\ell^2$ prediction error as reactive intrinsic reward (Stadie et al., 2015; Pathak et al., 2017) (**$\ell^2$ Error**); iii) PPO with VIME, i.e., $\text{IG}_\theta(\cdot)$ as a reactive intrinsic reward coupled with Bayesian Neural Network dynamics (Houthooft et al., 2016) (**VIME**); iv) PTS-BE with Deep Ensembles (Shyam et al., 2019); v) PTS-BE with SVGP (**PTS-BE-GP**); vi) PTS-BE with Deep Kernels (**PTS-BE-DK**).

We measure performance of the methods again via the cumulative fraction of visited states at each time step $t$, and the final fraction of coverage at episodic termination, which we set to be after 400 steps in this case, as the Markov chain of states is longer. PTS methods are left running for 10 steps initially in order to gather enough data to estimate $p_\theta(s_{t+1}|s_t, a_t)$. Results over 20 seeded replication of the experiment are reported in Figure 9's plots and table. Similarly to the unichain $L = 50$ states environment results presented in the main

paper, PTS methods consistently outperform the rest, as they require approximately 150 steps only to reach complete coverage of the environment and solve the task.

### D.3 Additional Details on the Medium Sized Maze

Finally, we report here a few extra information on the Maze environments. These are continuous control environments with $(\mathcal{S} \subseteq \mathbb{R}^4, \mathcal{A} \subseteq \mathbb{R}^2)$. The reward function is sparse, as the agents receive $r_t^e = 1$ only when they reach the objective, otherwise they receive $r_t^e = 0$. The environments are 2D, and their state space $\mathcal{S}$ features $x$ and $y$ coordinates of the agent and $x$ and $y$ coordinates linear velocity of the agent. The continuous action space $\mathcal{A}$ instead include coordinate's linear force applicable to the $x$ and $y$ directions.

These environment are highly customizable in terms of their shape and size. We experiment with different shapes and sizes of increasing difficulty. Figure 10 above reports a visual 3D rendition of some possible maze configuration achieved by customizing the map structure.

