# OpenReview forum: "On Efficient Bayesian Exploration in Model-Based Reinforcement Learning"
_TMLR — Accepted by TMLR_

### Review · Reviewer_G9j7 · 2025-04-27

**Summary Of Contributions:**

This work addresses data-efficient exploration in reinforcement learning (RL), a well-recognized challenge in the field. Efficient exploration depends highly on the ability to distinguish between epistemic and aleatoric uncertainty, encouraging exploration only in regions where epistemic uncertainty is high.

The authors propose using information gain (IG) as an intrinsic reward signal, which, unlike many existing approaches, managed to target only the epistemic uncertainty and naturally diminishes to zero over time (Section 3.1). Convergence guarantees are established by leveraging results from prior literature on information gain (Sections 3.2 and 3.3). Inspired by research on GPs and BNNs, the authors develop scalable and computationally tractable approximations to expected information gain (EIG), making the method applicable to complex RL settings, including both classic on-policy/off-policy algorithms for continuous control and model-based RL (Sections 4 and 5). Through illustrative experiments, they highlight the importance of distinguishing epistemic from aleatoric uncertainty - particularly under heteroskedastic noise - and show the effectiveness of their EIG-based approach (Sections 4.1 and 5.1).

**Audience:**

Yes

**Broader Impact Concerns:**

I don't immediately see any concerns.

**Claims And Evidence:**

Yes

**Requested Changes:**

1. Relation to Visitation-Frequency-Based Approaches: You should discuss more explicitly how your method relates to visitation-frequency-based exploration strategies. Many RL algorithms with regret guarantees fall into this category, where the intrinsic reward relates to visitation counts and can be interpreted as a form of epistemic uncertainty. Importantly, such intrinsic rewards also diminish with visitation at a non-asymptotic rate.

2. Clarifying the Novelty Compared to Prior Work: It would strengthen the paper to more clearly highlight the novelty of your approach relative to prior works, e.g., Houthooft et al. (2016), which you mention right before Section 3.2. How does their use of information gain as an intrinsic reward compare to or inspire your method?

Formatting and Minor Corrections
1. Figure 4: Colors are hard to distinguish.
2. Page 3: In the Bellman equation, the $V$ on the RHS should involve $p_\tau$.
3. Page 7: Remove the "=" at the end of each equation line.
4. Appendix Reference: Directly cite specific appendix sections instead of saying "provided in the appendix."
5. Algorithm 1, Line 8: Does $\mathcal{D}_{\text{model}}$ includes all $k \in [K]$ trajectories?

**Strengths And Weaknesses:**

## Strength

The paper is very well written and was a pleasant read.

As elaborated in the Contributions, unlike many existing methods, the authors define information gain (IG) under the RL setting that explicitly capture epistemic uncertainty, encouraging exploration where it is present. The proposed IG signal diminishes over time (Sections 3.2, 3.3), is made computationally tractable and scalable (Section 4), and can be incorporated into a wide range of RL methods, including on-policy, off-policy, and model-based approaches (Sections 4 and 5). The experiments are carefully designed and support the claims (e.g., Figure 1, Section 5).

## Weakness

Visitation-frequency-based methods also explicitly model epistemic uncertainty, diminishing with non-asymptotic rates, but this connection is not discussed in depth.

The novelty compared to prior work could be more explicitly highlighted. While this paper proposes several new ideas, it also draws inspiration from multiple fields. Clarifying which parts are directly adapted, which are inspired, and how, would better showcase its novelty.

---

> ### Author Response · Authors · 2025-05-14
> **Authors' Response to Reviewer G9j7**
>
> We thank the reviewer for their useful feedback and positive comments about the clarity of exposition. We proceed by addressing comments about weaknesses.
>
> 1. **Relation to Visitation-Frequency-Based Approaches**
>
> We appreciate the reviewer’s suggestion and fully agree that visitation-frequency (count-based) exploration merits further explicit discussion. In the revised manuscript we have added a dedicated paragraph to Section 3.1 that: (a) reviews classical count-based methods (MBIE-EB, UCRL2); (b) explains how both exact counts and pseudo-counts measure Bayesian information gain, thereby serving as an explicit proxy for epistemic uncertainty; and (c) notes that the associated intrinsic reward decays at the non-asymptotic rate $\mathcal{O} (1 / \sqrt{N(s, a)})$, ensuring that exploration incentives vanish once the agent is confident.
>
> We now also clarify why count-based bonuses are harder to propagate inside model-predictive-control planners, whereas our forward-model approach naturally provides both multi-step predictions and uncertainty estimates along imagined trajectories. We believe this additional discussion addresses the reviewer’s concern and further strengthens the theoretical positioning of the paper.
>
> 2. **Clarifying the Novelty Compared to Prior Work**
>
> We thank the reviewer for prompting a sharper comparison with Variational Information Maximizing Exploration (VIME; Houthooft et al. 2016) and other related works.
>
> In the revised version of the manuscript, we have correspondingly re-formulated the “Related Work” and “Contributions” paragraphs in Section 1, and adjusted other related claims throughout the manuscript, to better contrast our work with VIME and later IG papers (See also our related response to Reviewer Y3kG).
>
> In short, we now highlight how: i) VIME computes a one-step IG bonus using a variational BNN and applies it inside a model-free policy-gradient learner. In contrast, and similarly to Shyam et al. (2019), we argue in favor of embedding (E)IG in a model-based planner (PTS-BE) that looks H steps ahead and jointly reasons about extrinsic reward and epistemic gain; ii) While VIME (nor Shyam et al., 2019) provides no formal analysis of bonus decay, we provide a formal study of IG (and value functions) convergence and contraction rates; iii) The PTS-BE planner framework we utilise features a dynamics model for both rewards and next state, such that one can complement extrinsic and intrinsic rewards prediction, and also capture epistemic uncertainty stemming from the reward model.
>
> **Minor Corrections**: We thank the reviewer for spotting these minor inconsistencies. In the uploaded revised version of the manuscript these have now been all addressed.

---

### Review · Reviewer_Y3kG · 2025-04-28

**Summary Of Contributions:**

- Propose a new intrinsic reward that dynamically targets epistemic uncertainty (actually, the IG/EIG intrinsic reward was already proposed [[Shyam et al., 2019][2]], right?).
- Show that the intrinsic reward converges to zero in probability if the learned model converges to the true model in probability.
- Characterize the rate of convergence to zero in the special case of Gaussian processes.
- Show empirically that the intrinsic reward targets the epistemic noise, by ignoring the aleatoric noise in the case of heteroskedastic noise.
- Propose to use the learned model for planning in a model-based method (PTS-BE).
- Show that it works better than just using the learned model for the intrinsic rewards with a model-free method.
- Show that Gaussian Processes and Deep Kernels, which are more computationally costly, can show limited gains with respect to Deep Ensemble models.

[2]: https://proceedings.mlr.press/v97/shyam19a.html

**Audience:**

Yes

**Claims And Evidence:**

Yes

**Requested Changes:**

- If the intrinsic reward is not new (see weakness 1), please adjust your claims. Your abstract, introduction and conclusion really seem to imply that EIG is introduced for the first time in this paper. Generally speaking, I would appreciate the paper to be a little more precise on its contributions. I found difficult understanding what was existing in the literature and what was new.
- I would appreciate the ablation study "PTS" (see weakness 3) and a discussion of the source of performance of the algorithm (is it the model-based planning routine, or the intrinsic reward?). That said, I acknowledge that it is intersting that your intrinsic reward provides a model of the dynamics "for free".

**Strengths And Weaknesses:**

Strengths:
1. The aforementioned contributions seem interesting to me (although I am not sure all of them are contributions of this paper).
2. The experiments are well-designed and the results are interesting.

Weaknesses:
1. The relation with prior works is not clear (I acknowledge that I am not familiar with the Bayesian exploration literature).
    - It seems that this intrinsic reward is not new [[Latyshev & Panov, 2025][1]; [Shyam et al., 2019][2]]. This is not clearly stated in the related work section. Could you clarify how you differ from this line of work? Is your contribution to prove interesting properties for these existing intrinsic rewards using information-theoretic arguments?
    - It seems that model-based planning approaches were already used in conjonction with such model-based intrinsic rewards. This is not clear in your related work section. Could you clarify how your planning algorithm (PTS-BE) differs from previous model-based approaches? [[Shyam et al., 2019][2]; [Sekar et al., 2020][3]; [Mendonca et al., 2021][4]]
2. There seem to be a clash between the presented framework (Bayesian-adaptive MDP) and the method.
    - The BAMDP seem to provide samples from the posterior, while the algorithm seem to maintain the full posterior (and not to sample from this posterior), at least in the case of Gaussian processes. Could you better link the actual dynamic models update to $p(f | D)$ at line 14 in Algorithm 1 with the BAMDP's posterior distribution and its samples? I also think it could be interesting to better explain how the three methods (GP, DK, DE) compute $p(f | D)$ (or sample $f \sim p(\cdot | D)$) from the dataset.
3. The best methods are PTS-BE-{GP,DK,DE}, which suggests that most of the gain comes from the usage of a model-based planning approach.
    - I would be interested to see the ablation "PTS", a model-based approach with PPO, which would correspond to Algorithm 1 without intrinsic rewards.

Minor:
- Many equations have equality signs both at the end and at the beginning of lines (e.g., equations 6).
- Line 9 in Algorithm 9, there is a typo: $r_t^e$ instead of $r_{t+j}^e$.

[1]: https://link.springer.com/article/10.3103/S0147688224700370
[2]: https://proceedings.mlr.press/v97/shyam19a.html
[3]: https://proceedings.mlr.press/v119/sekar20a.html
[4]: https://proceedings.neurips.cc/paper/2021/hash/cc4af25fa9d2d5c953496579b75f6f6c-Abstract.html

---

> ### Author Response · Authors · 2025-05-14
> **Authors' Response to Reviewer Y3kG**
>
> We thank the reviewer for the time spent reviewing our submission and for their thoughtful feedback. We proceed by addressing the weaknesses.
>
> 1. **Relation to Prior Works**
>
> We appreciate the reviewer’s request for a clearer comparison and have substantially revised the “Related Work” and “Contributions” parts of our manuscript accordingly. In particular we have better detailed in these revised sections how:
>
> - IG/EIG is obviously not new; it has first a long-standing body of literature rooted in Bayesian Optimal Experimental Design (Lindley, 1956). It has been also proposed as an intrinsic reward in other works, such as the already cited VIME (Houthooft et al. 2016) and MAX (Shyam, 2019). The difference of our approach with VIME specifically is that VIME computes a one-step IG bonus using a variational BNN (model specific) and applies it inside a model-free policy-gradient learner. Similarly to MAX (Shyam et al., 2019), we argue in favor of embedding (E)IG in a model-based planner that looks H steps ahead and jointly reasons about extrinsic reward and epistemic gain. The PTS-BE framework simply generalizes MAX, which specifically advocates for the use of Deep Ensemble dynamics models, to be agnostic of the Bayesian model of the dynamics (GP, DK, Deep Ensemble, etc) - and to feature a model for the rewards as well, in order to incorporate both extrinsic and intrinsic rewards into the planner (as the objective $r^e + \eta EIG$) if one desires to do so and capture epistemic uncertainty stemming also from the reward model.
>
> - To the best of our knowledge, none of the relevant prior works (Houthooft et al. 2016, Shyam et al., 2019, Latyshev & Panov 2025) provide a formal proof of the desirable properties of these bonuses (Props 3.1 - 3.3) in terms of convergence and contraction rates that quantify sample complexity.
>
> In addition to the above improvements to Section 1, we have also adjusted our claims in other relevant parts of the manuscript.
>
> 2. **Clash between BAMDP and PTS algorithm**
>
> We thank the reviewer for pointing out the need to clarify, and ensure consistency, between our definition of BAMDP and the PTS algorithm.
>
> We have since identified and corrected aspects of the BAMDP definition that may have misled readers. In particular, the phrasing “$θ ∼ p(θ \mid (s_{t-1}, a_{t-1}, s_t))$” in Definition 2.1 could inadvertently suggest that the agent draws a single parameter sample from the posterior. In the revised manuscript, we have rephrased this as: “the current parameter belief given by the posterior $p(θ \mid (s_{t-1}, a_{t-1}, s_t))$”, emphasizing that the full belief posterior is maintained.
>
> Furthermore, in Algorithm 1, Line 14, we now explicitly clarify that this step corresponds to the BAMDP belief update: $p(f \mid D_t) \leftarrow p(f \mid D_{t-1})$. The nature of the belief posterior $p(f \mid D_t)$ depends on the choice of dynamics model:
>
> - GPs maintain a full multivariate Gaussian posterior $p(f \mid D)$, using (s, a) as inputs.
>
> - DKs maintain a multivariate Gaussian posterior using latent input representations $\phi(s, a)$. The marginalization is identical to the full GP, but performed in a latent space $\mathcal{H}$.
>
> - Stochastic Variational GPs and DKs (SVGP/SVDK) maintain a Gaussian posterior using a subset of representative inducing points in either (s, a) or $\phi(s, a)$, respectively. The marginalization is identical to the GP or DK, but carried out using only the inducing points.
>
> - Deep Ensembles (DEs) approximate the full belief posterior via an empirical mixture of $M$ probabilistic neural nets. Marginalization is done via Monte Carlo averaging: $\hat{p}(f \mid D) = \frac{1}{M} \sum_{m=1}^M \delta_{f_{\theta_m}}$.
>
> This posterior marginalization is used when rolling out imagined trajectories to solve the BAMDP. Using posterior samples from $p(f \mid D)$ instead would correspond to a stochastic approximation. Both are admissible BAMDP solutions, but the sampling-based approach generally incurs higher variance.
>
> We have made relevant improvements to Definition 2.1, Algorithm 1, and Section 4.2 to clarify how the belief posterior is handled and how its realization depends on the chosen dynamics model.
>
> 3. **PTS Ablation Study**
>
> Following the helpful recommendation we have now added a new baseline - “PTS-SAC” - to the final set of Maze Exploration Environments (Section 5.3 and Figure 5). This variant corresponds exactly to Algorithm 1 without the intrinsic reward term, isolating the contribution of planning alone. We use SAC as the policy learner (instead of PPO) to keep comparisons consistent with other baselines. PTS-SAC alone does not advance performance by a significant amount, confirming that the substantial gain stems from the synergy between the IG intrinsic rewards and the “planning-to-explore” approach, rather than planning alone.
>
> **Minor Comments**: We thank the reviewer for spotting these inconsistencies. We have amended these in the revised version of the manuscript.

---

> > ### Comment · Reviewer_Y3kG · 2025-05-15
> > **Thank you for your responses and modifications**
> >
> > Dear Authors,
> >
> > Thank you for your responses and modifications.
> >
> > My concerns and misunderstandings have been addressed by your modifications, and I think that the contributions are clearer now. I also appreciate that the posterior models better match your BAMDP formalization, and it is nice to see that the PTS-SAC baseline is performing significantly worse than PTS-BE.
> >
> > I just have one followup question out of curiosity. If I understood correctly your new discussion in paragraph "Relation to Visitation-Frequency Bonuses", such (vanishing) model-based bonuses do not present direct advantages with respect to (vanishing) visitation-based bonuses, except from the indirect benefit of having a learned dynamics model for free (which is shown very useful in practice). Is there a way to discuss how the visitation-based decay rate relates to your contraction rate for GPs? Or are these very different convergence characterization that do not permit a direct comparison?
> >
> > Kind regards,
> > Reviewer Y3kG

---

> > > ### Author Response · Authors · 2025-05-18
> > > **Authors' Response to Follow-Up Question**
> > >
> > > Dear Reviewer Y3kG,
> > >
> > > Thank you for the quick reply. We are glad the revisions to the manuscript have successfully addressed your concerns.
> > >
> > > Regarding your follow-up question about the relationship between Visitation-Frequency bonuses and IG convergence rates: this is an interesting observation that allows us to further discuss some points. The two rates (the standard $\mathcal{O} (N(s, a)^{-1/2})$ and $\mathcal{O}(n^{-1/(2 + |X|/\alpha)})$, respectively) arise in rather different statistical setups, so a one-to-one mapping is tricky. However, it is still useful to illustrate how they relate. In a visitation-based setup, each state-action pair $(s,a)$ is treated as an independent “cell”. The exploration bonus is proportional to $N(s,a)^{−1/2}$, where $N(s,a)$ is a local visit counter. This setting implicitly assumes a tabular (finite) space or, at the very least, that a good discretisation is possible. In our analysis of IG-based bonuses instead, we assume $(s,a)$ lives in a continuous domain where the dynamics function $f$ is characterized by some smoothness of order $\alpha$. The GP posterior - and therefore the IG bonus - contracts at a minimax rate that depends on the ratio between the input dimensionality and the smoothness $\alpha$. This result is particularly useful in continuous (s,a) spaces - where a good discretization can be hard, even via density models -, in that it ties the decay of the IG bonus to the complexity of the dynamics function $f$.
> > >
> > > Also, we emphasize that whether model-based IG bonuses offer advantages over visitation-based ones ultimately depends on the assumptions made about the structure of the $(s,a)$ space. In particular, in continuous domains, model-based approaches can provide statistical benefits when the dynamics exhibit exploitable and well-learnable structure (e.g., smoothness), whereas count-based methods hinge on how well the underlying density model or discretisation approximates the space. In that sense, the IG contraction rates we present offer a principled "guide" in setups where pseudo-count approximations may be less tractable or harder to interpret.
> > >
> > > We hope this adequately addresses your follow-up question, although this connection might merit further discussion. We are happy to include this further discussion on how our contraction results differ from visitation-based ones to the paper, at the end of Section 3.3.
> > >
> > > Kindest Regards,
> > >
> > > The Authors

---

> > > > ### Comment · Reviewer_Y3kG · 2025-05-20
> > > > **Thank you for your answer**
> > > >
> > > > Dear Author,
> > > >
> > > > Thank you for your interesting answer. I do not have an opinion on whether this should or should not be included in the paper.
> > > >
> > > > Best regards, \
> > > > Reviewer Y3kG

---

> > > > > ### Author Response · Authors · 2025-05-20
> > > > > **Final Thanks for Insightful Discussion**
> > > > >
> > > > > Dear Reviewer Y3kG,
> > > > >
> > > > > We would like to thank you once again for the insightful comments and for prompting an engaging discussion on the relationship between visitation-based and IG-based convergence rates.
> > > > >
> > > > > We will include a very brief note at the end of Section 3.3 that may help clarify to the reader the different assumptions and, more generally, how the respective convergence behaviors relate.
> > > > >
> > > > > Kindest Regards,
> > > > >
> > > > > The Authors

---

### Review · Reviewer_EccY · 2025-05-05

**Summary Of Contributions:**

This paper proposes a new class of exploration bonuses aimed at improving exploration efficiency in reinforcement learning. These bonuses are designed to approximate epistemic information gain and gradually converge to zero as the environment becomes fully explored. Building on this idea, the authors introduce an algorithm called *Predictive Trajectory Sampling with Bayesian Exploration (PTS-BE)*, which combines model-based planning with the proposed bonuses. Experiments on several sparse-reward and pure exploration environments show that PTS-BE outperforms a range of baselines.

**Audience:**

Yes

**Claims And Evidence:**

No

**Requested Changes:**

* Some notations in Section 4 are confusing and could be clarified. For example, in \( p_\theta(s_{t+1} \mid s_t, a_t; \theta) \), what is the intended meaning of \(\theta\) appearing both in the subscript and after the semicolon?
* In the experiments in Section 5.1, does the random policy eventually fully explore the environment? What is the coverage achieved?
* In Figure 3 (middle), why is performance of $H(\cdot)$ in the *Homosk* setting lower than in *Heterosk*? Intuitively, the heteroskedastic case should be more challenging.
* The implementation of the "BE" baseline in Section 5.1.1 is not entirely clear. How is the reward augmentation carried out? It would be helpful if the authors could elaborate on this.

**Strengths And Weaknesses:**

**Strengths**

* The paper is mostly clearly written.
* The proposed exploration bonus is theoretically motivated and grounded.

**Weaknesses**

* While the theoretical foundation is appealing, the paper does not demonstrate that the method remains effective in more complex or high-dimensional environments.

---

> ### Author Response · Authors · 2025-05-14
> **Authors' Response to Reviewer EccY**
>
> We thank the reviewer for the time spent reviewing our submission. We proceed by addressing comments about weaknesses and requested changes.
>
> ### Weaknesses
>
> 1. **Effectiveness in Complex Environments**
>
> We thank the reviewer for the comment. We agree that evaluating performance in higher dimensional domains remains important future work in this area.
>
> While our paper offers a primarily theoretical contribution, establishing the essential groundwork for information-theoretic exploration bonuses, we are currently in the process of running the PTS-BE-SAC pipeline on other Gymnasium Robotics benchmarks with sparse rewards (e.g. FetchReach, FetchSlide, and related tasks - https://robotics.farama.org/). These experiments will empirically validate how our approach scales to continuous, high-dimensional control problems, and we will report their outcomes in the camera-ready version.
>
> ### Requested Changes
>
> 1. **Confusing Notation in Section 4**
>
> We thank the reviewer for spotting these inconsistencies in Section 4. In the revised version of the manuscript we have correspondingly corrected our notation resulting in improved readability. In particular, $p_{\theta} (s_{t+1} | s_t, a_t; \theta)$ contains a typographic error that overloads the \theta term. The equation has now been changed to: $p (s_{t+1} | s_t, a_t) = \int p (s_{t+1} | s_t, a_t; \theta) p(\theta | s_t, a_t) \, d\theta$. We have amended this issue in all previous occurrences throughout Section 4.
>
>
> 2. **Random Policy Coverage in Section 5.1 Experiments**
>
> We agree that a direct comparison to the random policy would give clearer insights on the results of Section 5.1. We have now added a random policy baseline to both the homoskedastic and heteroskedastic Mountain Car experiments in Section 5.1 (in the updated Figure 3). The random policy achieves under 40% state‐space coverage in 1.000 steps, under 70% in 5.000 steps, and about 90% only after 10.000 steps. Note that in none of these instances does it solve the environment’s task. By contrast, PTS-BE-IG solves the task 18/20 times in the heteroskedastic setting and 17/20 in the homoskedastic one, in fewer than 1000 steps; while PTS-BE-Entropy solves it only 8/20 and 10/20 times, respectively, within the same budget, demonstrating once again the superiority of (E)IG as an intrinsic reward signal. We’ve added a brief discussion of these results to Section 5.1 of the revised manuscript.
>
> 3. **Performance of PTS-BE-Entropy in the homoskedastic vs heteroskedastic case**
>
> While it is true that the PTS-BE-Entropy curve in the heteroskedastic case lies slightly above the one in the homoskedastic case, their error bands still substantially overlap. Entropy-based PTS-BE essentially rewards noisy states, which may result in a slightly higher fraction of visited states that are driven primarily by stochasticity. Notice how in the BE case the exact opposite occurs—that is, the BE-Entropy heteroskedastic curve lies slightly below the homoskedastic one. We also note that in both heteroskedastic and homoskedastic scenarios, PTS-BE-Entropy methods exhibit higher variance, as indicated by significantly larger error bands compared to PTS-BE-IG methods. This emphasizes how the planning-to-explore strategy with information gain rewards is a more stable choice. We have added a comment about this in the revised version of Section 5.1.
>
> 4. **Unclear Implementation of the "BE" baseline in Section 5.1.1**
>
> We thank the reviewer for the opportunity to clarify this point. We have now improved the implementation description in Section 5.1.1 of the revised manuscript as-per the following explanation:
>
> The “BE” specification refers to a model-free solver in which, before each policy update, we augment the collected transitions’ extrinsic rewards with the chosen intrinsic bonus (either Entropy or EIG).
>
> Concretely, in Section 5.1.1 we use PPO as the base solver, so at each update we form a dataset $D_T = \\{ (s_i, a_i, r^{e}_i, s'_i) \\}^{T}_t $ of environment interactions, then replace each reward $r^e_t$ with $r_t = r^e_t + \eta r^i_t$, where $r^i_t$ is the intrinsic bonus for that transition (Entropy or EIG) and $\eta$ its scaling factor. We then perform the usual PPO update, minimizing the clipped surrogate loss.

---

### Author Response · Authors · 2025-05-14
**Authors' General Comment**

We are thankful to the reviewers for the time spent going through our submission and for their insightful comments, as we believe these have significantly contributed to improving the paper. We are happy to share that we have uploaded a revised version of the manuscript that incorporates their suggestions (changes are marked in blue to facilitate cross-referencing).

In summary, we have addressed the following points:

- **Positioning and Novelty**. We have re-written the Related Work and Contributions sections (and few other relevant parts in the manuscript) to clarify how our contribution is positioned with respect to previous relevant works

- **Additional Discussion & Clarifications**:

     - *Count-based exploration*. Section 3.1 now contains a dedicated paragraph reviewing MBIE-EB, UCRL2 and pseudo-count methods, and explains why their bonuses are less amenable to propagation inside model-predictive-control planners.

    - *BAMDP formulation*. Definition 2.1, Algorithm 1 and Section 4.2 have been re-written to make the posterior belief update and its model-specific realisations (GP, DK, SVGP/SVDK, Deep Ensembles) explicit.

    - *Notation, equations and figures*. We corrected typos, clarified overloaded symbols and improved colour palettes for accessibility.

- **New Experiments and Ablations**. We added the PTS-SAC baseline (planning without intrinsic reward) to isolate the benefit of the information-gain bonus, and a random-policy baseline to the Mountain-Car study to benchmark state-space coverage.

- **Minor fixes**. All formatting issues (equation alignment, appendix pointers, algorithm details) flagged by the reviewers have been resolved.

We hope we have adequately responded to all the concerns regarding the paper, and remain available to engage with the reviewers during the rest of the discussion period, to respond to any outstanding queries.

Kindest Regards,

The Authors

---

### Decision · Action_Editor_tmyh · 2025-06-05

**Recommendation:** Accept with minor revision

**Additional Comments:**

The reviewers (mostly) agree that the paper is interesting and the main concerns have been addressed through back-and-forth discussions with the authors. The paper is deemed "easy to understand and interesting" while it "provided new insights into the field". Reviewers have further praised the comparison of various posterior approximators and the analysis on the impact of generating trajectories with the learned model.

However, I think a few changes shall be considered for the final version of the paper:
- The first sentence of the abstract could be rephrased: "developing a principled information-theoretic approach to intrinsic motivation" sounds like a new exploration bonus is derived, whereas the paper actually covers analysis and implementation of an existing bonus;
- The contributions section may further highlight the comparison of tractable posterior approximators for IG as a core contribution, just like it is done in the abstract;
- The authors have promised further experiments on Gymnasium Robotics tasks, which I agree with reviewers they would strengthen the paper contribution.

**Audience:**

Yes

**Audience Explanation:**

The paper may attract the attention of a reinforcement learning audience, with special attention to intrinsic motivation methods.

**Claims And Evidence:**

Yes

**Claims Explanation:**

The contributions claimed in the abstract mostly capture what is provided in the paper.

---

> ### Author Response · Authors · 2025-07-01
> **Decision and Camera Ready Comments**
>
> Dear Action Editor,
>
> We are very pleased to hear that the paper has been accepted. Once again, we would like to thank both the reviewers and AE for the valuable feedback and thoughtful discussions that have helped improve the quality and clarity of the manuscript.
>
> We have now uploaded the updated camera-ready version, in which we have addressed the final round of comments. In particular:
>
> 1. We have rephrased parts of the abstract to clarify that the paper studies, but does not derive, intrinsic exploration bonuses;
>
> 2. We have slightly revised the contributions paragraph to explicitly highlight the comparison of tractable posterior approximators for information gain as a core contribution;
>
> 3. We have added an additional set of experiments at the end of Section 5, featuring the Ant Maze environment from the Gymnasium Robotics suite, which presents higher-dimensional state and action spaces and a more complex sparse-reward exploration task.
>
> In case anything relevant is missing, please do not hesitate to contact us. Thank you again for your support throughout the process.
>
> Best Regards,
>
> The Authors